# Changes in March mean snow water equivalent since the mid-twentieth century and the contributing factors in reanalyses and CMIP6 climate models

Jouni Räisänen[1]

[1] Institute for Atmospheric and Earth System Research / Physics, University of Helsinki, P.O. Box 64, FI-00014 University of Helsinki, Finland

*Correspondence to*: Jouni Räisänen (jouni.raisanen@helsinki.fi)

**Abstract.** Trends in March mean snow water equivalent (SWE) in the Northern Hemisphere are attributed to changes in three main factors: total precipitation ($P$), fraction of precipitation as snowfall ($F$), and fraction of accumulated snowfall remaining on ground ($G$). This trend attribution is repeated for two reanalyses (ERA5-Land from March 1951 to 2022 and MERRA2 from 1981 to 2022) and simulations by 22 climate models from the 6th phase of the Coupled Model Intercomparison Project (CMIP6). The results reveal a decrease in SWE in most of the Northern Hemisphere, as decreases in $F$ and $G$ dominate over mostly positive trends in $P$. However, there is spatial variability in both the magnitude and sign of these trends. There is substantial variation between the individual CMIP6 models, but the agreement between the CMIP6 multi-model mean and ERA5-Land is reasonable for both the area means and the geographical distribution of the trends from 1951 to 2022, with a spatial correlation of 0.51 for the total SWE trend. The agreement for the trends from 1981 to 2022 is worse, probably partly due to internal climate variability but also due to the overestimation of the recent warming in the CMIP6 models. Over this shorter period for which ERA5-Land can be compared with MERRA2, there are also marked trend differences between these two reanalyses. However, the SWE decreases associated with reduced snowfall fraction ($F$) are more consistent between the different data sets than the trends resulting from changes in $P$ and $G$.

## 1 Introduction

Simulations of greenhouse gas induced climate change by global climate models feature both warming and, in the northern mid-to-high latitudes in winter, an increase in precipitation (Lee et al., 2021). These changes have opposing effects on snowpack. An increase in precipitation, if acting alone, would increase the amount of snowfall and snow on ground. However, an increase in mean temperature favours the occurrence of above-zero at the expense of below-zero temperatures, particularly where and when the mean temperature is relatively close to the freezing point (de Vries et al., 2013; Räisänen, 2016). Therefore, a smaller fraction of precipitation falls as snow (Krasting et al., 2013; Kapnick and Delworth, 2013; Räisänen, 2016) and the snowpack is reduced by more frequent and intense melt events during the winter (Musselman et al., 2021). The net effect of these changes in climate model simulations is a shortening of the snow season (Brown and Mote, 2009; Zhu et al., 2021) and

a decrease in the snow water equivalent (SWE) in most areas (Mudryk et al., 2020). However, in the coldest regions such as eastern Siberia and northern Canada, the increase in total precipitation tends to dominate and thus leads to an increase in SWE at the height of the snow season (Räisänen, 2008; Brown and Mote, 2009).

Changes in snow cover and snow amount, as characterised either by snow depth or SWE, are important from climatological, hydrological, ecological, and many other points of view. Climatologically, snow cover increases the surface albedo, thus acting as an amplifier of temperature changes (Thackeray and Fletcher, 2016). It also isolates the air from the heat storage in the ground, which moderates the winter cooling of the soil but further decreases the surface air temperature, over and above the albedo effect (Vavrus, 2007). Hydrologically, the water stored in the snowpack reduces river flows in winter but increases

them during and after the spring snowmelt, which also replenishes the soil moisture in the beginning of the growing season (Vavrus, 2007; Li et al., 2017; Qi et al., 2020). A thick enough snowpack also provides shelter for rodents and other small animals against the winter cold, along with many other beneficial ecological impacts (Petty et al., 2015). For all these reasons, it is of interest to ask whether recently observed changes in real-world snow conditions have followed the expectations from climate model simulations.


Earlier research on changes in snow conditions in the past few decades gives two key insights. The first is a general albeit non-uniform decrease in the extent and amount of snow. For example, Kunkel et al. (2016) reported a predominantly decreasing trend in winter maximum snow depth from winter 1960/61 to 2014/15 in North America and Europe, based on station observations in the Global Historical Climatology Network daily data set. Pulliainen et al. (2020) evaluated trends in snow

mass in 1980-2018 using the Global Snow Monitoring for Climate Research (GlobSnow) v3.0 analysis. Focusing on non-mountainous areas north of 40° N, they found a statistically significant decreasing trend in March mean snow mass in North America (best estimate: -4 % decade$^{-1}$) but a near-zero trend in Eurasia. Mudryk et al. (2020) derived consensus estimates of variations in Northern Hemisphere snow extent and snow mass in years 1981-2018, basing the former (latter) on the average of six (four) data sets produced using various methodologies and sources of observations. These estimates showed a decreasing

trend in snow extent and snow mass throughout the year, but especially from November to June. Using observations for the years 1981-2010, Mudryk et al. (2017) found the monthly mean snow cover extent in the extratropical Northern Hemisphere land areas to decrease throughout the snow season by $(1.9 \pm 0.9) \times 10^6$ km$^2$ for each 1 °C increase in the mean temperature in the same month and area.

The second key message from earlier research is the need to consider observational uncertainty (Mudryk et al., 2017; Mortimer et al., 2020). A fundamental reason of this is the insufficiency of direct measurements. As much of the Northern Hemisphere snow cover resides in sparsely populated areas poorly covered by station and snow course observations, a hemispheric view on snow conditions requires satellite remote sensing and / or numerical modelling. The National Oceanic and Atmospheric Administration (NOAA) has produced snow charts based on manual analysis of primarily visible light satellite images since

the year 1966 (Robinson et al., 2012). However, Mudryk et al. (2017, 2020) question the homogeneity of this record, since its 1981-to-2010 trends disagree with several other data sets. Notably, the NOAA snow charts suggest a pronounced increase of snow extent in October that appears physically incompatible with the simultaneous warming.

Remote sensing of SWE is more challenging than that of snow extent. SWE estimates based on the attenuation of microwave radiation within the snowpack exist (Kelly, 2009; Tedesco and Jeyaratnam, 2016), but these stand-alone microwave products still agree less well with in-situ observations than gridded SWE data sets based on alternative techniques (Mortimer et al., 2020). The latter include GlobSnow (Takala et al., 2011; Pulliainen et al., 2020), several reanalyses that assimilate available observations to a simulation by a weather prediction or climate model, and SWE data sets produced by a land surface model forced by observed or analysed time series of near-surface meteorological variables. The GlobSnow technique, which combines information from microwave measurements and in-situ snow depth observations, appears currently as one of the most promising approaches (Pulliainen et al., 2020). However, it is not feasible in mountainous areas where in-situ observations are sparse and surface conditions are strongly variable. Reanalyses and land surface models produce spatially complete SWE simulations, in some cases with high spatial and temporal resolution (e.g., hourly data at 9 km resolution for ERA5-Land (Muñoz Sabater et al., 2021)). On the other hand, model biases and limitations in the observational input result in a large spread between the SWE estimates from different analysis products (Mudryk et al., 2015; Mortimer et al., 2020). Mudryk et al. (2015) found more than a factor of 1.5 range even in the Northern Hemisphere total winter peak snow mass among the five datasets that they evaluated (their Fig. 3a). More importantly for the climate change perspective, real trends in snow conditions may be confounded with temporal inhomogeneity in the input observations. This problem is especially acute for reanalyses where new types of satellite observations are introduced to the assimilation data stream with time. For example, Mortimer et al. (2020) report a downward discontinuity in the European Centre for Medium Range Weather Forecasts (ECMWF) ERA5 reanalysis (Hersbach et al., 2020) snow mass around the year 2004, when the assimilation of satellite-derived binary snow-no-snow estimates began.

This paper focuses on recent trends in SWE in the Northern Hemisphere and the mechanisms that have contributed to them. In addition to exploring how SWE has changed, we proceed one step further and attribute this change to the three multiplicative factors that determine SWE (Räisänen, 2008; 2021a): total precipitation, the fraction of precipitation falling as snow (the snowfall fraction), and the fraction of accumulated snowfall that has not melted away and thus remains on ground at a given time of the winter season (the snow-on-ground fraction). The primary questions that the paper aims to address are thus

1. How has SWE changed?
2. How have the changes in total precipitation, snowfall fraction and snow-on-ground fraction individually contributed to the SWE change?
3. Are the changes in climate model simulations consistent with those in the real world?

In practice, the answers to these questions are complicated by both observational uncertainty, differences between climate models, and internal climate variability. This raises three auxiliary questions:

4.  How well do we know how SWE has changed, concerning both the total SWE change and the contributions of the three multiplicative factors?

5.  How much do the historical SWE change and the contributions of the three multiplicative factors vary between different climate model simulations?

6.  How much of the differences in SWE change between climate model simulations, and between climate models and the real world, may be explained by internal variability?

Throughout the paper, the focus will be on SWE in March, when the Northern Hemisphere snow mass is the largest (Pulliainen et al., 2020). In the interest of simplicity, SWE changes will be characterized by least-squares linear trends. For the reasons discussed in Section 3, these trends are calculated for two periods of time, from March 1951 to 2022 and from 1981 to 2022.

A key finding of this research is a reasonable agreement between the ERA5-Land reanalysis and the CMIP6 (6th phase of the
Coupled Model Intercomparison Project; Eyring et al., 2016) models on the March mean SWE trends and their contributing factors in the period 1951-2022 (Section 5) but a worse agreement between various observational data sets with both each other and the CMIP6 models on the trends from 1981 to 2022 (Section 6).

**2 Data sets**

The diagnostic framework which represents SWE using total precipitation, snowfall fraction, and snow-on-ground fraction
(Section 3) requires three variables: total precipitation, snowfall, and SWE. Total precipitation is reported regularly at thousands of stations, but without separating the contributions of rainfall and snowfall. Furthermore, although networks of SWE measurement exist, for example, in Canada (Brown et al., 2019), Russia (Bulygina et al., 2011) and Finland (Haberkorn, 2019), their coverage is relatively limited (Mortimer et al., 2020). Therefore, the current study relies on reanalysis data sets in describing the "observed" evolution of precipitation, snowfall, and SWE.

The first criterion in selecting the reanalyses was temporal coverage for at least the past four decades up to the present day, to get a statistically meaningful and up-to-date view on the currently ongoing SWE change. Second, reanalyses that directly assimilate snow-related land surface variables were discarded, because data assimilation may create a mismatch between SWE and the atmospheric forcing (temperature, precipitation, etc.) that regulates snowfall and snowmelt. Two global reanalyses

meeting these criteria were found: ERA5-Land (Muñoz Sabater et al., 2021) and MERRA2, the Modern-Era Retrospective analysis for Research and Applications, Version 2 (Gelaro et al. 2017; Reichle et al., 2017).

ERA5-Land (hereafter ERA5L) is a land-only rerun of the ERA5 reanalysis, produced by forcing the H-TESSEL land surface model (Balsamo et al., 2009; Dutra et al., 2010) with ERA5 meteorological output downscaled to 9 km resolution. ERA5L is available from year 1950 to the present. MERRA2 is an atmosphere-land reanalysis produced by version 5.12.4 of the Goddard Earth Observing System atmospheric data assimilation system. It is available from year 1980 to the present in a $0.5° \times 0.625°$ latitude-longitude grid. Neither ERA5L nor MERRA2 assimilates observations of land surface variables, leaving the evolution of SWE solely determined by the land surface model used and the meteorological forcing provided to it.

To assess the uncertainty in ERA5L and MERRA2 and to help the interpretation of climate trends, the auxiliary data sets summarized in Table 1 were used. Of the two variants of the GlobSnow v3.0 SWE data set (Pulliainen et al., 2020; Luojus et al., 2021), the bias-corrected one was chosen. These corrections are based on comparison with snow course measurements, and they improve the SWE estimates especially in areas with thick snow, where the non-corrected data systematically underestimate SWE due to the saturation of the microwave signal when SWE exceeds ca. 150 mm (Pulliainen et al., 2020). Currently, GlobSnow is being superseded by the European Space Agency Snow Climate Change Initiative SWE data set (Mortimer et al., 2022), but bias corrections have not yet been implemented to it.

**Table 1.** Auxiliary observational data sets. T = temperature; P = precipitation; SLP = sea level pressure. CRU = Climatic Research Unit Time Series v4.06 (Harris et al., 2020); GPCC = Global Precipitation Climatology Centre (Schneider et al., 2022); GPCP = Global Precipitation Climatology Project Version 2.3 (Adler et al., 2018); ERA5 = ECMWF ERA5 reanalysis (Hersbach et al., 2020); GlobSnow = GlobSnow v3.0 (Luojus et al., 2021). Further details on the availability and processing of these data sets are given in Appendix A.

| Acronym | Variable | Years of record | Native resolution | Purpose of use |
|---------|----------|-----------------|-------------------|----------------|
| CRU | T | 1901-2021 | 0.5° | assessment of reanalysis T trends (Section 7.3) |
| | P | 1901-2021 | 0.5° | assessment of reanalysis P trends (Section 7.3) |
| GPCC | P | 1891-2020 | 0.5° | assessment of reanalysis P trends (Section 7.3) |
| GPCP | P | 1979-2022 | 2.5° | assessment of reanalysis P trends (Section 7.3) |
| ERA5 | T | 1940-2022 | 0.25° | global mean temperature trend (Section 7.3) |
| | SLP | 1940-2022 | 0.25° | trend in atmospheric circulation (Section 7.3) |
| GlobSnow | SWE | 1980-2018 | 25 km | model-independent estimate of SWE (Sections 4 and 6) |

Climate model simulations from CMIP6 were also used, concatenating historical simulations for the years 1950-2014 with simulations for the Shared Socioeconomical Pathways "middle of the road" 2-4.5 scenario (Fricko et al., 2017) for years 2015-2022. The analyzed simulations form two groups (Table 2):

1. A 22-model ensemble was formed using one realization per model (r1i1p1f1 or r1i1p1f2 depending on data availability). The variation within this ensemble includes the combined effects of model differences and internal variability. The mean value of these 22 simulations is referred to as the multi-model mean (MMM).

2. For the five models with the largest number of realizations with different initial conditions (28 to 50 depending on model), all these realizations were used to isolate the variance caused by internal variability without the confounding effect of model differences (Section 7.2).

**Table 2.** CMIP6 models used in the study. Model acronyms follow https://esgf-node.llnl.gov/search/cmip6/ (ESGF, 2023). The atmospheric horizontal resolution is given in degrees latitude × degrees longitude. Main realization refers to the realization used in calculating the multi-model mean and the inter-model variance. $N$ = Number of realizations used. $\Delta T_{51-22}$ and $\Delta T_{81-22}$ give the Total Snow Area November-to-March mean temperature trends in these periods in winters 1951-2022 (°C (71 yr)$^{-1}$) and 1981-2022 (°C (41 yr)$^{-1}$). The model and ensemble numbers are used in Figs. 4-5.

| Model | Ensemble | Model acronym | Atmospheric resolution | Main realization | $N$ | $\Delta T_{51-22}$ | $\Delta T_{81-22}$ |
|---|---|---|---|---|---|---|---|
| 1 | | ACCESS-CM2 | 1.25° × 1.875° | r1i1p1f1 | 1 | 2.3 | 2.4 |
| 2 | 1 | ACCESS-ESM1-5 | 1.25° × 1.875° | r1i1p1f1 | 40 | 2.9 | 2.5 |
| 3 | | BCC-CSM2-MR | 1.125° × 1.125° | r1i1p1f1 | 1 | 1.9 | 2.0 |
| 4 | 2 | CanESM5 | 2.813° × 2.813° | r1i1p1f1 | 50 | 3.2 | 2.7 |
| 5 | | CNRM-CM6-1 | 1.406° × 1.406° | r1i1p1f2 | 1 | 3.3 | 1.4 |
| 6 | | CNRM-CM6-1-HR | 0.5° × 0.5° | r1i1p1f2 | 1 | 2.7 | 2.4 |
| 7 | | CNRM-ESM2-1 | 1.406° × 1.406° | r1i1p1f2 | 1 | 3.3 | 2.2 |
| 8 | | EC-Earth3-CC | 0.703° × 0.703° | r1i1p1f1 | 1 | 3.2 | 2.5 |
| 9 | | EC-Earth3 | 0.703° × 0.703° | r1i1p1f1 | 1 | 4.1 | 4.3 |
| 10 | | EC-Earth3-Veg | 0.703° × 0.703° | r1i1p1f1 | 1 | 3.4 | 2.5 |
| 11 | | EC-Earth3-Veg-LR | 1.125° × 1.125° | r1i1p1f1 | 1 | 2.6 | 1.9 |
| 12 | | GFDL-ESM4 | 1.0° × 1.0° | r1i1p1f1 | 1 | 1.3 | 1.8 |
| 13 | | GISS-E2-1-G | 2.0° × 2.5° | r1i1p1f2 | 1 | 1.5 | 1.9 |
| 14 | | GISS-E2-1-H | 2.0° × 2.5° | r1i1p1f2 | 1 | 2.7 | 2.3 |
| 15 | | IPSL-CM6A-LR | 1.26° × 2.5° | r1i1p1f1 | 1 | 2.6 | 2.4 |
| 16 | 3 | MIROC6 | 1.406° × 1.406° | r1i1p1f1 | 33 | 2.0 | 1.8 |
| 17 | 4 | MIROC-ES2L | 2.813° × 2.813° | r1i1p1f2 | 28 | 2.4 | 2.2 |
| 18 | | MPI-ESM1-2-HR | 0.938° × 0.938° | r1i1p1f1 | 1 | 1.9 | 1.7 |
| 19 | 5 | MPI-ESM1-2-LR | 1.25° × 1.875° | r1i1p1f1 | 30 | 2.1 | 1.6 |
| 20 | | MRI-ESM2-0 | 1.125° × 1.125° | r1i1p1f1 | 1 | 2.1 | 2.5 |
| 21 | | NorESM2-MM | 0.938° × 1.25° | r1i1p1f1 | 1 | 2.4 | 2.0 |
| 22 | | UKESM1-0-LL | 1.25° × 1.875° | r1i1p1f2 | 1 | 2.7 | 3.1 |

All the observational data sets and the CMIP6 simulations were interpolated to a common 2.5° × 2.5° latitude-longitude grid using first-order conservative remapping (Jones, 1999). This leads to a loss of local information particularly for the data sets with the highest resolution, such as ERA5L. However, the 2.5° × 2.5° grid is sufficient for a large-scale analysis, and the trend decomposition results in this grid are nearly independent of whether the grid remapping is done before or after the

decomposition (Eqs. (1)-(2) in Section 3). See Fig. B1 for an illustration of the resolution dependence of the 1951-to-2022 trends in ERA5L in the Scandinavian area.

## 3. Methods

Our diagnostic framework follows Räisänen (2008, 2021a). Only three variables are needed from a reanalysis or a model simulation: monthly means of SWE, snowfall, and total precipitation ($P$). The monthly snowfall is first rewritten as $FP$, where $F$ is the fraction of precipitation that falls as snow. SWE in month $t$ then becomes

$$SWE = G \int_{t_0}^{t} FP dt' \tag{1}$$

where $t_0$ denotes the beginning of the snow year, here fixed to August. The snow-on-ground fraction $G$ is diagnosed by dividing the monthly mean of $SWE$ by the time integral of snowfall ($FP$). In evaluating the latter, August ($t_0$) and month $t$ (in this study, March) are given half-weight because the SWE data used in the analysis represent monthly means rather than end-of-month values. To make Eq. (1) also applicable in areas where snow cover regularly or sporadically survives to the late summer, we subtract the August mean SWE from the left-hand-side, thus focusing on the seasonal component of SWE. For reference, in ERA5L about 7 % of the Total Snow Area (red and yellow shading in the bottom-left panel of Fig. 1) has non-negligible (> 5 mm) time mean August SWE in the 2.5° × 2.5° grid, largely in mountainous and Artic areas. For MERRA2, this fraction is only 1 %.

To analyse how variations in $G$, $F$, and $P$ have contributed to the trends in SWE, a two-step procedure is followed. First, the same quasi-linearization as in Räisänen (2021a) is used to decompose the SWE anomalies in individual winters. The monthly mean values of $X = SWE$, $G$, $F$ and $P$ over the whole analysis period are denoted as $X_1$, whereas their values in an individual winter are denoted as $X_2$. By further defining $\bar{X} = (X_1 + X_2)/2$ and $\Delta X = X_2 - X_1$, one obtains

$$\Delta SWE(t) = \underbrace{\bar{G} \int_{t_0}^{t} \bar{F} \Delta P dt'}_{\Delta SWE(\Delta P)} + \underbrace{\bar{G} \int_{t_0}^{t} \Delta F \bar{P} dt'}_{\Delta SWE(\Delta F)} + \underbrace{\Delta G \int_{t_0}^{t} \bar{F} \bar{P} dt}_{\Delta SWE(\Delta G)} + \underbrace{\frac{1}{4} \Delta G \int_{t_0}^{t} \Delta F \Delta P dt'}_{\Delta SWE(NL)} \tag{2}$$

Thus, the anomaly in SWE is decomposed to contributions from the total precipitation ($\Delta P$), snowfall fraction ($\Delta F$), and snow-on-ground fraction anomalies ($\Delta G$), plus a non-linear term that is typically two orders of magnitude smaller than the others. However, there is an implicit non-linearity in the coefficients $\bar{G}$, $\bar{F}$, and $\bar{P}$ in (2) since, for example, $\bar{G} = G_1 + \Delta G/2 \neq G_1$.

Second, least-square linear trends in $\Delta SWE$ and its four components are calculated. This is repeated for two periods, winters 1951 to 2022 and 1981 to 2022. Thanks to its length, the former period maximises the signal-to-noise ratio between forced climate change and internal variability. However, MERRA2 only covers the latter period. The total SWE trend is additionally calculated for winters 1981 to 2018, to allow an unbiased comparison between GlobSnow and the other data sets.

Where area mean values for different data sets or spatial correlations between them are reported, this is done for either the *Total Snow Area* or the *GlobSnow Area*. The former includes those 2.5° × 2.5° land grid boxes at latitudes 30-80° N where the climatological mean SWE in winters 1981-2022 (as averaged over ERA5L and MERRA2) exceeds 5 mm at least in one calendar month. However, Greenland is excluded. The GlobSnow Area is a subset of the Total Snow Area, covering about 81% of it. It excludes those 2.5° × 2.5° grid boxes in which more than half of the GlobSnow data in their original finer grid were missing, meaning mountainous areas and latitudes south of 40° N. These two averaging areas are shown in the bottom-left panel of Fig. 1.

When representing trends in maps, stippling is used as a broad indicator of robustness. Reanalysis trends are stippled where they exceed the 5-95 % range for trends generated by white noise interannual variability. For CMIP6, stippling is used where the MMM trend exceeds the inter-model standard deviation.

## 4. Average snow climate

In this section, the March mean SWE climate in winters 1981-2022 is compared between the observational data sets and the CMIP6 models, and the factors contributing to it are analysed. SWE is shown in the fourth column of Fig. 1 and the three multiplicative factors that contribute to it in the first three columns. For this figure and Tables 3-4, Eq. (1) is rewritten as

$$SWE = GF^*P^* \tag{3}$$

where

$$P^* = \int_{t_0}^{t} P dt' \tag{4}$$

is the total precipitation integrated from August to March (with half-weight for August and March) and $F^*$ is the snowfall fraction for the same period.

The large-scale geographical patterns are similar for ERA5L, MERRA2 and the CMIP6 MMM, and they show physically expected features. The snowfall and snow-on-ground fractions $F^*$ and $G$ increase from warm to cold climates: from south to north, from the relatively mild western Europe towards the interior and eastern parts of Eurasia, and with increasing elevation. This makes the distribution of March mean SWE rather different from that of the August-to-March total precipitation ($P^*$). Yet the latter also matters. For example, the relatively modest SWE in eastern Siberia is due to small total precipitation, while the SWE in some mountainous regions (notably the west coast of Canada) is amplified by very large total precipitation.

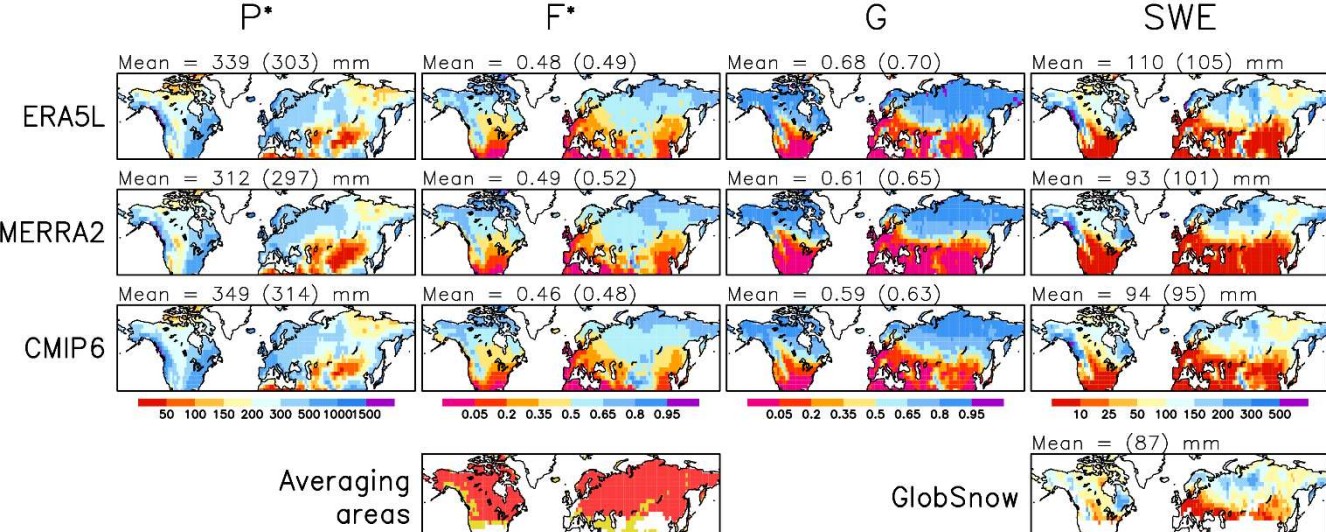

**Figure 1.** Mean values of mid-August to mid-March total precipitation and snowfall fraction (columns 1-2), and snow-on-ground fraction and SWE (columns 3-4) in March in winters 1981-2022 in ERA5-Land, MERRA2 and the CMIP6 MMM. For SWE, the GlobSnow estimate for March 1981-2018 is shown in the bottom-right panel. The numerical values in the headers show the area means, where the mean values for the snowfall fraction ($F*$) are weighted by the total precipitation ($P*$) and those for the snow-on-ground fraction ($G$) by the accumulated snowfall ($F*P*$). The mean values without parentheses are for the Total Snow Area (including both the red and the yellow shading in the bottom-left panel) and those in parentheses for the GlobSnow Area (red shading in the bottom-left panel).

The similarity of geographical patterns suggested by Fig. 1 is confirmed by the high spatial correlations between ERA5L, MERRA2 and the CMIP6 MMM (Table 3). For each of the three factors $P*$, $F*$ and $G$, these correlations vary from 0.88 to 0.96 in the Total Snow Area and are even higher (0.93-0.97) in the GlobSnow Area. Even for SWE, which is affected by the differences in all of $P*$, $F*$ and $G$, correlations close to 0.9 are found, except between ERA5L and MERRA in the Total Snow Area ($r = 0.79$). However, the correlations of *SWE* between GlobSnow and the other three data sets (0.78-0.82) are lower than those among these three. Thus, the GlobSnow *SWE* distribution differs more from ERA5L, MERRA2 and the CMIP6 MMM than the latter three differ from each other.

**Table 3.** Spatial correlation of the fields shown in Fig. 1 between different data sets. The values without (within) parentheses are for the Total Snow Area (the GlobSnow Area).

|                    | $P*$        | $F*$        | $G$         | SWE         |
|--------------------|-------------|-------------|-------------|-------------|
| ERA5L vs. MERRA2   | 0.93 (0.95) | 0.95 (0.97) | 0.88 (0.94) | 0.79 (0.89) |
| ERA5L vs. CMIP6    | 0.96 (0.96) | 0.95 (0.96) | 0.93 (0.95) | 0.87 (0.92) |
| MERRA2 vs. CMIP6   | 0.90 (0.93) | 0.93 (0.94) | 0.95 (0.95) | 0.88 (0.93) |
| ERA5L vs. GlobSnow |             |             |             | (0.78)      |
| MERRA2 vs. GlobSnow |            |             |             | (0.79)      |
| CMIP6 vs. GlobSnow |             |             |             | (0.82)      |

Despite the pattern similarity, there are quantitative differences between the data sets. As a first-order indicator of these, the area means of $P^*$, $F^*$, $G$ and $SWE$ are given in the map headers of Fig. 1, for both the Total Snow Area and the GlobSnow Area, using weighting that preserves the identity (3) for these area means (see the figure caption). The mean SWE in ERA5L exceeds that in MERRA2, but the difference is much larger in the Total Snow than the GlobSnow Area. The average SWE in the mountainous, mostly relatively low-latitude regions that are excluded from the GlobSnow Area is 129 mm in ERA5L but only 65 mm in MERRA2. Such a large difference between the two reanalyses is remarkable, although it might partly reflect the higher resolution, and therefore steeper orography, in ERA5L (~0.1°) than in MERRA2 (0.5° × 0.625°). Regardless of the averaging area, the snow-on-ground fraction $G$ is also larger in ERA5L than MERRA2, while the average snowfall fraction $F^*$ is slightly larger in MERRA2.

The CMIP6 22-model mean SWE in the Total Snow Area is close to MERRA2 but 15 % below ERA5L; in the GlobSnow Area it is also below MERRA2. The average precipitation in the CMIP6 models exceeds both ERA5L and MERRA2, but this is compensated by lower mean values of $F^*$ and $G$ (third row in Fig. 1). On the other hand, the average CMIP6 SWE exceeds the GlobSnow estimate (bottom-right corner in Fig. 1) by nearly 10 %. Kouki et al. (2022) also used GlobSnow as their main observational data set, finding an average ~15 % overestimate of March mean SWE for a larger set of 38 CMIP6 models (their Fig. 2b). Both the different sets of models and the inclusion of mountainous areas (where GlobSnow was replaced by other observational estimates) by Kouki et al. (2022) may contribute to this slight difference. Using linear regression, Kouki et al. (2022) attributed the overestimate of simulated SWE in February to too large November-to-January precipitation in the CMIP6 models. Although they made this regression analysis for February rather than March mean SWE, this result is in line with the CMIP6 MMM overestimate of area mean $P^*$ suggested by Fig. 1.

Assuming that GlobSnow and the other observational estimates used by Kouki et al. (2022) are correct, the average SWE is too large in both the CMIP6 MMM, MERRA2, and (especially) ERA5L. A comparison of GlobSnow with three other bias-corrected estimates of the total snow mass in Northern Hemisphere non-alpine areas (Table 1 of Pulliainen et al., 2020) supports this assessment: all the four estimates are within 7.4 %, GlobSnow being the highest. However, SWE in mountainous areas is known less well and may be severely underestimated in many gridded analyses (Snauffer et al., 2016; Wrzesien et al., 2018). Despite the higher mean SWE in ERA5L than in MERRA2 and GlobSnow, Muñoz-Sabater et al. (2021) found ERA5L to underestimate SWE by ~50 % at the five Earth System Model – Snow Model Intercomparison Project (Krinner et al., 2018) alpine reference sites.

**Table 4.** Total Snow Area mean values of $P^*$, $F^*$, $G$ and SWE for March 1981-2022, and the area mean trend in March mean SWE and its three main components (Eq. 2) in years 1951-2022 and 1981-2022 (in parentheses) in the 22 CMIP6 models, ERA5L and MERRA2. $F^*$ and $G$ are non-dimensional; the other values are in mm. The largest and smallest values in the CMIP6 ensemble are underlined.

| Model acronym | Climate 1981-2022 | | | | Trend 1951-2022 (1981-2022) | | | |
|---|---|---|---|---|---|---|---|---|
| | $P^*$ | $F^*$ | $G$ | SWE | $\Delta SWE(\Delta P)$ | $\Delta SWE(\Delta F)$ | $\Delta SWE(\Delta G)$ | $\Delta SWE$ |
| ACCESS-CM2 | 327 | 0.49 | 0.53 | 84 | 8.6 (11.5) | -9.2 (-10.5) | -0.9 (-0.6) | -1.4 (0.4) |
| ACCESS-ESM1-5 | 366 | 0.43 | 0.49 | 76 | 8.2 (9.5) | -11.8 (-10.8) | -2.8 (-3.4) | -6.3 (-4.6) |
| BCC-CSM2-MR | 339 | 0.47 | 0.63 | 101 | 7.4 (9.5) | -9.9 (-12.4) | -3.8 (-4.8) | -6.3 (-7.6) |
| CanESM5 | 324 | 0.44 | 0.60 | 86 | 13.1 (9.9) | -12.4 (-11.1) | -8.4 (-6.4) | -7.7 (-7.6) |
| CNRM-CM6-1 | 362 | 0.43 | 0.63 | 99 | 13.1 (8.9) | -11.7 (-6.3) | -10.5 (-5.6) | -9.0 (-2.9) |
| CNRM-CM6-1-HR | 357 | 0.48 | 0.65 | 111 | 12.9 (9.8) | -12.6 (-10.2) | -7.1 (-6.5) | -6.7 (-6.8) |
| CNRM-ESM2-1 | 369 | 0.42 | 0.62 | 96 | 11.7 (7.4) | -11.9 (-8.2) | -10.4 (-7.3) | -10.5 (-8.1) |
| EC-Earth3-CC | 327 | 0.43 | 0.66 | 94 | 12.3 (12.6) | -12.7 (-11.8) | -7.4 (-6.3) | -7.8 (-5.4) |
| EC-Earth3 | 319 | 0.46 | 0.71 | 105 | 19.8 (21.3) | -23.3 (-23.0) | -6.1 (-6.7) | -9.5 (-8.4) |
| EC-Earth3-Veg | 320 | 0.45 | 0.70 | 101 | 13.1 (10.5) | -15.9 (-12.1) | -5.5 (-6.1) | -8.2 (-7.6) |
| EC-Earth3-Veg-LR | 315 | 0.45 | 0.70 | 99 | 8.5 (8.3) | -13.6 (-10.9) | -4.7 (-5.0) | -9.7 (-7.6) |
| GFDL-ESM4 | 363 | 0.46 | 0.49 | 81 | 2.3 (6.4) | -4.7 (-8.3) | -4.0 (-4.9) | -6.4 (-6.8) |
| GISS-E2-1-G | 377 | 0.54 | 0.65 | 132 | 12.1 (7.4) | -8.9 (-9.2) | -2.3 (-0.8) | 1.0 (-2.6) |
| GISS-E2-1-H | 381 | 0.52 | 0.67 | 133 | 14.1 (13.0) | -11.8 (-10.8) | -6.4 (-5.2) | -4.0 (-3.0) |
| IPSL-CM6A-LR | 378 | 0.52 | 0.57 | 112 | 10.6 (12.3) | -11.2 (-10.1) | -6.1 (-6.9) | -6.6 (-4.5) |
| MIROC6 | 358 | 0.42 | 0.60 | 89 | 6.2 (5.3) | -8.2 (-7.4) | -7.2 (-5.2) | -9.1 (-7.3) |
| MIROC-ES2L | 366 | 0.40 | 0.46 | 67 | 9.0 (10.0) | -5.7 (-5.8) | -9.7 (-10.8) | -6.4 (-6.5) |
| MPI-ESM1-2-HR | 360 | 0.44 | 0.41 | 66 | 6.0 (5.0) | -5.6 (-6.7) | -6.7 (-5.2) | -6.3 (-6.9) |
| MPI-ESM1-2-LR | 357 | 0.44 | 0.45 | 70 | 6.1 (5.2) | -6.3 (-4.4) | -5.9 (-5.4) | -6.0 (-4.6) |
| MRI-ESM2-0 | 388 | 0.42 | 0.61 | 99 | 7.5 (12.1) | -11.8 (-14.8) | -6.6 (-5.9) | -10.9 (-8.5) |
| NorESM2-MM | 302 | 0.50 | 0.75 | 113 | 14.0 (11.5) | -11.3 (-13.1) | -4.4 (-2.9) | -1.7 (-4.5) |
| UKESM1-0-LL | 317 | 0.47 | 0.39 | 59 | 8.8 (10.7) | -8.6 (-11.2) | -3.0 (-5.2) | -2.8 (-5.7) |
| Mean | 349 | 0.46 | 0.59 | 94 | 10.3 (9.9) | -10.9 (-10.4) | -5.9 (-5.3) | -6.5 (-5.8) |
| Median | 358 | 0.45 | 0.62 | 98 | 9.8 (9.9) | -11.5 (-10.7) | -6.0 (-5.3) | -6.5 (-6.1) |
| Standard deviation | 25 | 0.04 | 0.10 | 20 | 3.8 (3.5) | 4.0 (3.8) | 2.5 (2.1) | 3.1 (2.3) |
| ERA5L | 339 | 0.48 | 0.68 | 110 | 7.1 (5.2) | -8.0 (-8.2) | -7.4 (-3.9) | -8.3 (-6.9) |
| MERRA2 | 312 | 0.49 | 0.61 | 93 | (2.9) | (-5.5) | (-0.5) | (-3.1) |

 The CMIP6 MMM hides considerable inter-model variability (Table 4, columns 1-4). Among the 22 models, the average Total Snow Area March mean SWE varies from 59 mm in UKESM1-0-LL to 133 mm GISS-E2-1-H. The mean values of $P^*$, $F^*$, $G$ and SWE in both ERA5L and MERRA2 fall within the range of the CMIP6 simulations, although there is only one model (NorESM2-MM) in which the August-to-March total precipitation is smaller than in MERRA2.

 Table 4 also shows that the inter-model differences in $G$ are in relative terms larger than those in $P^*$ and $F^*$. This is perhaps unsurprising, since $G$ may be affected by a multitude of factors. As defined by Eq. (1), $G$ reflects the balance between the source (accumulated snowfall) and sinks (snowmelt plus sublimation) of snow. The accumulated snowfall depends on both

the amount and phase of precipitation, whereas snowmelt and sublimation are ultimately determined by the amount of energy that the land surface model allocates to them. The latter, in turn, is constrained by the downward solar and thermal radiation reaching the surface, the exchange of sensible and latent heat between the land surface and the atmospheric models, the description of the surface albedo and emissivity, and the use or release of energy associated with temperature changes within the snow-ground-vegetation system. As many of these processes are described differently in different land surface models, it is perhaps unsurprising that the simulated SWE may vary substantially even between land surface models that share the same atmospheric forcing (Mudryk et al., 2015). A more detailed understanding of the causes of variation of $G$ within the CMIP6 ensemble is an important target for future research.

## 5. Trends from winter 1951 to 2022

We next study the changes in SWE and their decomposition using Eq. (2), starting from the trends from March 1951 to 2022 in this section and continuing with the shorter 1981-to-2022 period in Section 6. Comparison of the 1951-2022 March trends between ERA5L and the CMIP6 MMM reveals similar large-scale features but differences in details (Figure 2). Increases in total precipitation have acted to increase SWE in most of the extratropical Northern Hemisphere (column 1) but this has been compensated by reduced snowfall fraction (column 2). The trends in $\Delta SWE(\Delta G)$, representing the changes in the snow-on-ground fraction, are also mostly negative but geographically variable (column 3). This term is the most negative in mid-latitude North America and in a zone extending from eastern Europe to southern Scandinavia, where the main snowmelt season is ongoing in March and has been advanced by rising spring temperatures. Conversely, the snow-on-ground fraction has locally increased at higher latitudes in North America and in parts of Siberia in ERA5L, although this increase is rarely statistically significant (note the lack of stippling in Fig. 2). It also increases slightly in broadly the same areas in the CMIP6 MMM. Although warming is generally expected to enhance snowmelt, this effect is modest where the mean temperature in March and in the preceding winter months is well below zero, so that above-zero temperatures remain uncommon despite the warming (Räisänen, 2008). Furthermore, where the accumulated winter snowfall increases, the snow-on-ground fraction also increases if the relative increase in snowmelt is smaller than that in snowfall.

In most areas, the decreases in the snowfall and snow-on-ground fractions dominate over the increase in total precipitation, leading to a decrease in March mean SWE in both ERA5L and CMIP6 (Fig. 2, column 4). Yet there are increases in Alaska, northern Canada, and Siberia. The SWE trends in ERA5L and the CMIP6 MMM have similar large-scale distributions, but the trends in ERA5L are patchier. The trends differ in sign, for example, in northern Fennoscandia (decrease in CMIP6 but increase in ERA5L due to a larger increase in total precipitation) and in easternmost Siberia (increase in CMIP6 but decrease in ERA5L, again reflecting different precipitation trends). At the west coast of North America at ca. 45-50° N, decreases in precipitation make the SWE trend more strongly negative in ERA5L than in CMIP6.

The spatial correlation between ERA5L and the CMIP6 MMM is 0.45 for $\Delta SWE(\Delta P)$, 0.75 for $\Delta SWE(\Delta F)$, 0.58 for

$\Delta SWE(\Delta G)$, and 0.51 for the $SWE$ trend in the Total Snow Area (Table 5). These values are distinctly lower than the mean

climate correlations reported in Table 3 but higher than the corresponding correlations for the 1981-2022 trends (to be

discussed in Section 6).

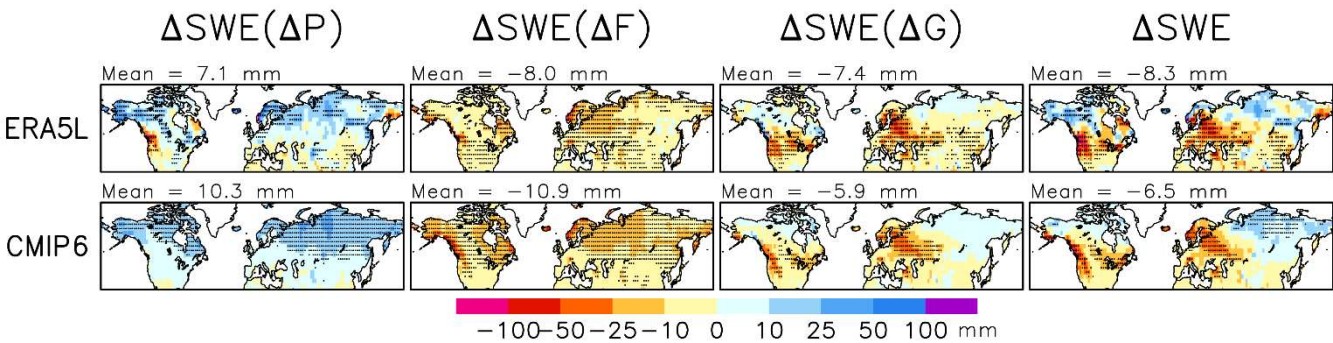

**Figure 2.** Trend in March mean SWE from 1951 to 2022 (mm (71 yr)$^{-1}$) (column 4) and the contributions to it from changes in total precipitation (column 1), snowfall fraction (column 2) and snow-on-ground fraction (column 3) in ERA5L and CMIP6 MMM. The area means for the Total Snow Area are given in the headers.

**Table 5.** Spatial correlation of the trend in March mean SWE and its contributing terms (Eq. 2) between different data sets. SWE-EUR and SWE-NAM refer to SWE trends in Eurasia and North America, respectively. The values without (within) parentheses represent the Total
340 Snow Area (GlobSnow Area).

| Years | | $\Delta SWE(\Delta P)$ | $\Delta SWE(\Delta F)$ | $\Delta SWE(\Delta G)$ | $SWE$ | $SWE$-EUR | $SWE$-NAM |
|---|---|---|---|---|---|---|---|
| 1951-2022 | ERA5L vs. CMIP6 | 0.45 | 0.75 | 0.58 | 0.51 | 0.59 | 0.49 |
| 1981-2022 | ERA5L vs. MERRA2 | 0.42 | 0.79 | 0.39 | 0.29 (0.48) | 0.52 (0.70) | 0.11 (0.15) |
| | ERA5L vs. CMIP6 | 0.35 | 0.61 | 0.27 | 0.09 (0.16) | 0.45 (0.37) | -0.34 (-0.12) |
| | MERRA2 vs. CMIP6 | 0.17 | 0.57 | 0.30 | 0.16 (0.12) | 0.23 (0.21) | 0.02 (-0.05) |
| 1981-2018 | ERA5L vs. GlobSnow | | | | (0.13) | (0.41) | (0.02) |
| | MERRA2 vs. GlobSnow | | | | (0.24) | (0.31) | (0.16) |
| | CMIP6 vs. GlobSnow | | | | (0.34) | (0.51) | (0.07) |

     The positive trend in $\Delta SWE(\Delta P)$ and the negative trend in $\Delta SWE(\Delta F)$ are both larger for the CMIP6 MMM than ERA5L, but these differences are dwarfed by the variation between the individual CMIP6 models (Table 4). The largest positive
$\Delta SWE(\Delta P)$ and negative $\Delta SWE(\Delta F)$ trends occur in EC-Earth3, which also stands out as the model with the largest winter warming (Table 2, column $\Delta T_{51-22}$). In one model (GISS-E2-1-G), the area mean SWE increases slightly in March.

## 6. Trends from winter 1981 to 2022

     The trends in March mean SWE from 1981 to 2022 and their contributing factors are shown in Fig. 3 for ERA5L, MERRA2, and the CMIP6 MMM. Additionally, the SWE trend in GlobSnow is given for the slightly shorter period 1981-2018. The
350 predominant sign of the trends is the same as for the 1951-to-2022 trends in Fig. 2: positive for $\Delta SWE(\Delta P)$ but negative for

$\Delta SWE(\Delta F)$, $\Delta SWE(\Delta G)$, and SWE. The CMIP6 MMM trends for 1981-2022 are also very similar in pattern to those for 1951-2022, but the ERA5L trends are not. The Total Snow Area spatial correlation between the SWE trends in these two periods is 0.97 for the CMIP6 MMM, but only 0.16 for ERA5L. For example, increases in SWE in northern Eurasia in ERA5L are much more widespread in 1951-2022 than in 1981-2022, except for easternmost Siberia, where SWE decreases in the former period but increases in the latter (top-right panels in Figs. 2 and 3). In western North America, where ERA5L indicates a pronounced SWE decrease in 1951-2022, the trend in 1981-2022 is far more subtle.

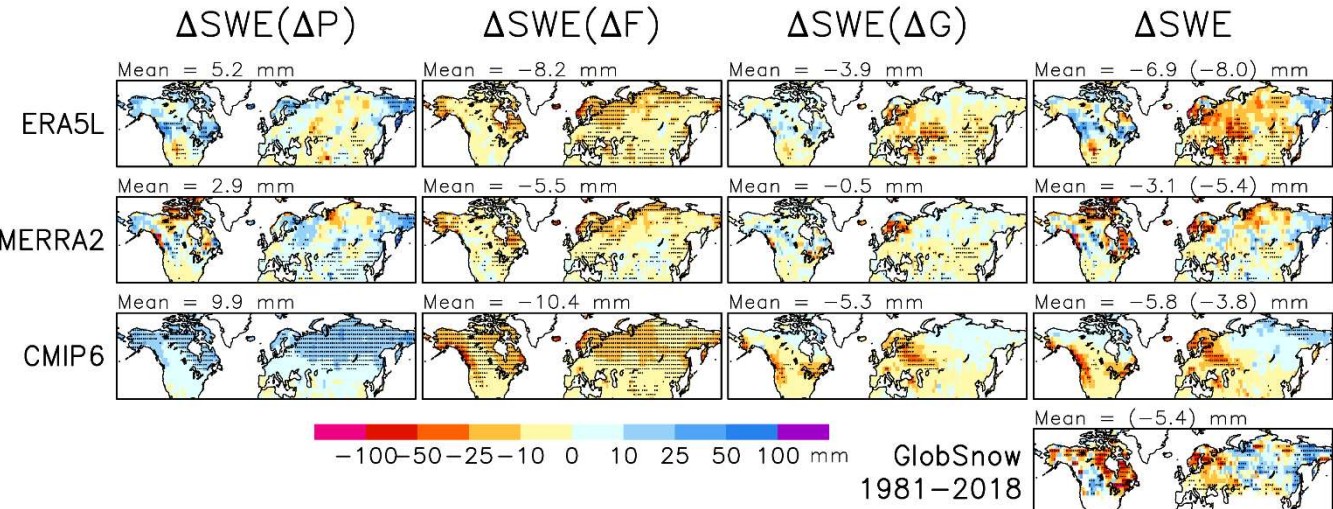

**Figure 3.** Rows 1-3: Trend in March mean SWE from 1981 to 2022 (mm (41 yr)$^{-1}$) (column 4) and the contributions to it from changes in total precipitation (column 1), snowfall fraction (column 2) and snow-on-ground fraction (column 3). Mean values over the Total Snow Area are given in the map headers. For SWE, the mean trend in the GlobSnow Area from 1981 to 2018 is also given (in parentheses). Row 4: SWE trend in GlobSnow from 1981 to 2018 (mm (37 yr)$^{-1}$).

The greater between-period dissimilarity of trends in ERA5L than in the CMIP6 MMM can be partly explained by internal variability (Section 7.2). However, comparison between ERA5L and MERRA2 points to observational uncertainty as another potentially important factor (rows 1-2 in Fig. 3 and row 2 in Table 5). The Total Snow Area spatial correlation between the 1981-to-2022 SWE trends in the two reanalyses is only 0.29, and the correlations for the individual terms of Eq. (2) are nearly as low, apart from $\Delta SWE(\Delta F)$ ($r = 0.79$). Thus, ERA5L and MERRA2 agree reasonably well on the SWE trend caused by changing snowfall fraction (which, as argued below, is a relatively straightforward response to warming), but much less well on the trends associated with changes in total precipitation and the snow-on-ground fraction. Furthermore, over the slightly shorter period 1981-2018 and in the GlobSnow Area, the spatial correlation between the ERA5L (MERRA2) and GlobSnow SWE trends is only 0.13 (0.24).

The CMIP6 MMM SWE trends in March 1981-2022 (or 1981-2018) are not well correlated with any of the observational data sets (Table 5). In particular, the correlation with ERA5L is much lower for the trend in 1981-2022 (0.09) than in 1951-2022

(0.51). The CMIP6-to-reanalysis spatial correlations for the individual terms in Eq. (2) are also modest for $\Delta SWE(\Delta P)$ and $\Delta SWE(\Delta G)$ but higher (~0.6) for $\Delta SWE(\Delta F)$.

The contribution to the SWE trend from changing snowfall fraction ($\Delta SWE(\Delta F)$) agrees better between ERA5L, MERRA2 and CMIP6 MMM than the other components or the March mean SWE trend in 1981-2022, and the same holds when comparing CMIP6 and ERA5L in 1951-2022 (Table 5). This is most likely (i) because the phase of precipitation is primarily (Auer, 1974) although not completely (Sims and Liu, 2015; Jennings et al., 2018) determined by temperature and (ii) because the observational uncertainty is smaller (Gulev et al., 2021) and the signal-to-noise ratio is higher (Räisänen, 2001; Hawkins and Sutton 2009, 2011; Lehner et al., 2020) for temperature than precipitation changes. Temperature also regulates snowmelt and thus affects the snow-on-ground fraction. However, this effect is less straightforward because of the confounding effect of precipitation changes (witnessed by the slightly positive $\Delta SWE(\Delta G)$ trends in Figs. 2 and 3 in some of the coldest areas) and probably also because of the complexity of modelling the snowmelt process.

Averaged over the Total Snow Area, the positive $\Delta SWE(\Delta P)$ trend and the negative $\Delta SWE(\Delta F)$ and $\Delta SWE(\Delta G)$ trends are all largest for the CMIP6 MMM and smallest for MERRA2, with ERA5L falling between these two (Fig. 3). The smallness of $\Delta SWE(\Delta P)$ in the two reanalyses is unusual relative to the inter-model variability, as the MERRA2 estimate is below and the ERA5L estimate close to the CMIP6 minimum (Table 4). The very mildly negative $\Delta SWE(\Delta G)$ trend in MERRA2 is also slightly outside of the CMIP6 range.

The CMIP6 MMM Total Snow area mean SWE trend in March 1981-2022 (-5.8 mm) falls between the trends in ERAL (-6.9 mm) and MERRA2 (-3.1 mm), whereas the corresponding trend in March 1981-2018 in the GlobSnow area (-3.8 mm) is slightly less negative than those in ERA5L, MERRA2, and GlobSnow (-5.4 to -8 mm). The Mudryk et al. (2020) consensus estimate of Northern Hemisphere March mean SWE trend in the same period suggests an even larger decrease (~9 mm by unit conversion from their Fig. 1c).

The SWE trends in the various data sets have higher spatial correlations in Eurasia than in North America (last two columns of Table 5). However, there is a striking discrepancy in both the Eurasian and North American area mean 1981-to-2022 and 1981-to-2018 SWE trends between ERA5L and the other data sets (Table B1). ERA5L suggests an increase in average SWE in North America and a major decrease in Eurasia, while MERRA2, GlobSnow, and the CMIP6 MMM all indicate larger decreases in North America than Eurasia. In particular, GlobSnow shows a near-zero SWE trend in Eurasia, but a 17 mm decrease from 1981 to 2018 in North America, in good agreement with Pulliainen et al. (2020). Mirroring these mean values, maps of the inter-data-set trend differences (Fig. B2) reveal a particularly pervasive difference between ERA5L and GlobSnow, with more negative trends in ERA5L in much of Eurasia but more positive trends in North America (Fig. B2c). Yet, in the longer 1951-to-2022 period, the SWE decrease in ERA5L is also slightly larger in North America than in Eurasia (Table B1).

**7. Discussion**

The results in the two previous sections reveal several common features in the SWE trend and its contributing factors between the CMIP6 MMM, ERA5L and MERRA2. However, many differences are also evident, particularly in the trends starting in winter 1981. This raises several questions.

1. Are the differences between the multi-model mean trends and the analysed trends compatible with the variation between
the individual model simulations? If not, this suggests a problem either in the analyses or in the reliability of the CMIP6 ensemble. Conversely, if the differences between the analysed and simulated trends are comparable with the inter-model differences, this supports the statistically indistinguishable ensemble paradigm (Annan and Hargreaves, 2010) in which model-simulated and real-world trends belong to the same statistical population. This question will be studied in Section 7.1.

2. There are two distinct causes for the inter-model differences in the simulated trends: differences in the models themselves (and in the details of the forcing applied), and internal climate variability. To study the likely importance of the latter, the variance of trends within five single-model initial condition ensembles is compared with the variance in the multi-model CMIP6 ensemble in Section 7.2.

3. In MERRA2, changes in total precipitation make a smaller positive contribution to the SWE trend since winter 1981 than
any of the models simulate, and the trend in ERA5L is also exceeded in 20 of the 22 models (Table 4). Similarly, the decrease in SWE due to reduced snowfall fraction is smaller in the reanalyses than in most models. To help understand these results, the trends in winter temperature and precipitation are studied and the potential causes of the model-to-reanalysis differences in them are explored in Section 7.3.

**7.1 Are the analysed and simulated trends consistent with the indistinguishable ensemble paradigm?**

The indistinguishable ensemble paradigm posits that climate changes in model simulations and in the real world should belong to the same statistical population (Annan and Hargreaves, 2010). The validity, or lack thereof, of this paradigm has important implications for projections of future climate, but it can only be tested for those changes that have already occurred. Therefore, combined model plus observation ensembles were formed by concatenating the 22-model CMIP6 ensemble with either
ERA5L, MERRA2, or GlobSnow. Then, the trends in each 23 members of this ensemble were compared with the mean of the other 22 members, using two statistics: the spatial correlation and the mean absolute difference (MAD). Again, we focus on trends in March in the Total Snow Area (or the GlobSnow Area for comparison with GlobSnow).

As an example, the March mean SWE trends in ERA5L in 1981-2022 are compared with the CMIP6 trends in Fig. 4. The
spatial correlation of the ERA5L trend with the mean of the 22 CMIP6 models is lower than the correlation between any single

CMIP6 model and the mean of the remaining 21 + 1 ensemble members (Fig. 4a). The MAD for ERA5L is the second highest (Fig. 4b). Thus, the ERA5L SWE trends in 1981-2022 appear unusual compared with the CMIP6 ensemble, although this conclusion is weaker for MAD than the spatial correlation.

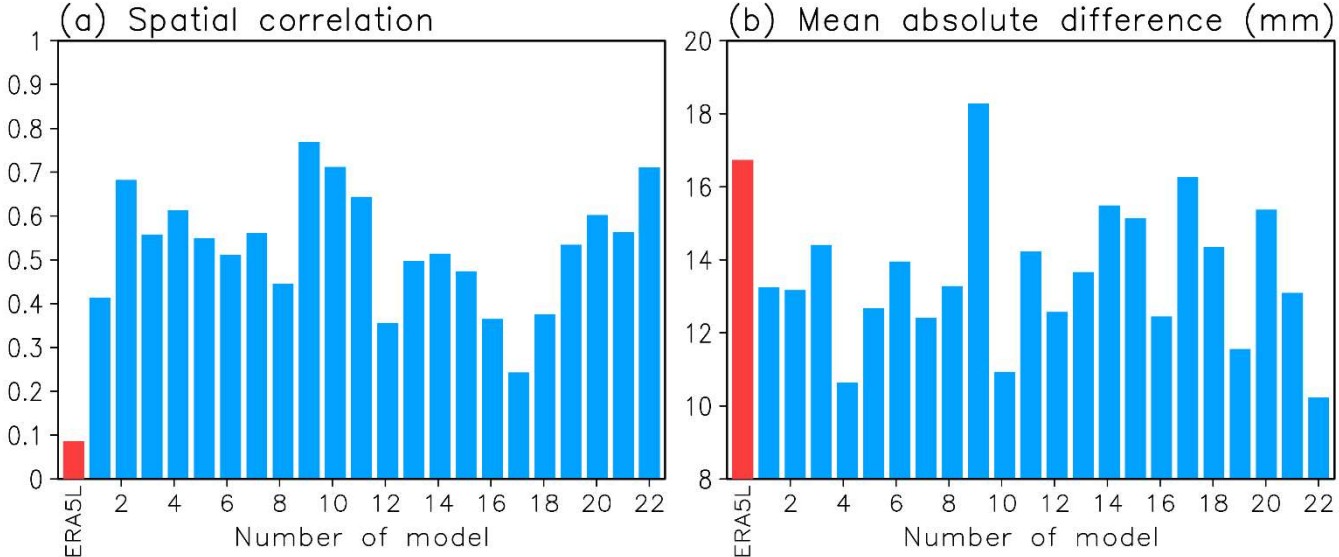

**Figure 4.** Consistency test results for comparison of 1981-to-2022 March mean SWE trends between ERA5L and the 22 CMIP6 models. (a) Spatial correlation between ERA5L (red) and each of the 22 models (blue) with the mean of the 22 other trend fields in the Total Snow Area. (b) As (a) but for the MAD from the mean of the 22 other trend fields.

The results for this and several other cases are summarized in Table 6. Just as in ERA5L, the 1981-to-2022 SWE trends in MERRA2 and the 1981-to-2018 trends in GlobSnow are near the outer edge of the CMIP6 distribution, as more divergent

trends are only found in 0-2 models depending on the statistic used. By contrast, the ERA5L SWE trends in March 1951-2022 do not stick out since a lower spatial correlation (higher MAD) is found for 5 (6) of the 22 CMIP6 models. Regarding the components of the SWE trend, the precipitation change contribution $\Delta SWE(\Delta P)$ in both ERA5L and MERRA2 appears unusual in comparison with the CMIP6 trends in 1981-2022. Any evidence of discrepancy in the other cases is weaker. However, there is a systematic difference between the correlation and the MAD measures, suggesting that the spatial patterns

of the reanalysis trends are more discordant with the CMIP6 ensemble than the magnitude of the trends.

**Table 6.** Number of CMIP6 models (out of 22) in which the simulated trends in March agree less well with the mean of the rest of the combined simulation plus analysis data set than the analyzed trends do, as measured by the spatial correlation (CORR) and the mean absolute difference (MAD). See the text for further explanation.

| Years | Analysis | $\Delta SWE(\Delta P)$ CORR | MAD | $\Delta SWE(\Delta F)$ CORR | MAD | $\Delta SWE(\Delta G)$ CORR | MAD | $SWE$ CORR | MAD |
|---|---|---|---|---|---|---|---|---|---|
| 1951-2022 | ERA5L | 3 | 6 | 10 | 13 | 12 | 18 | 5 | 6 |
| 1981-2022 | ERA5L | 2 | 1 | 4 | 11 | 2 | 15 | 0 | 1 |
| | MERRA2 | 0 | 1 | 4 | 8 | 2 | 16 | 0 | 2 |
| 1981-2018 | GlobSnow | | | | | | | 2 | 1 |

## 7.2 Effect of internal variability on the inter-model variation of the simulated trends

The variance between the 22 CMIP6 models in the March mean SWE trend and its three main components was calculated separately for each grid box and averaged over the Total Snow Area. The results are shown with the red bars in Fig. 5. The same was then repeated for each of the five single-model ensembles, each with 28-50 realizations of climate evolution starting from different initial conditions (Table 2). These variances are represented by the blue bars in Fig. 5.

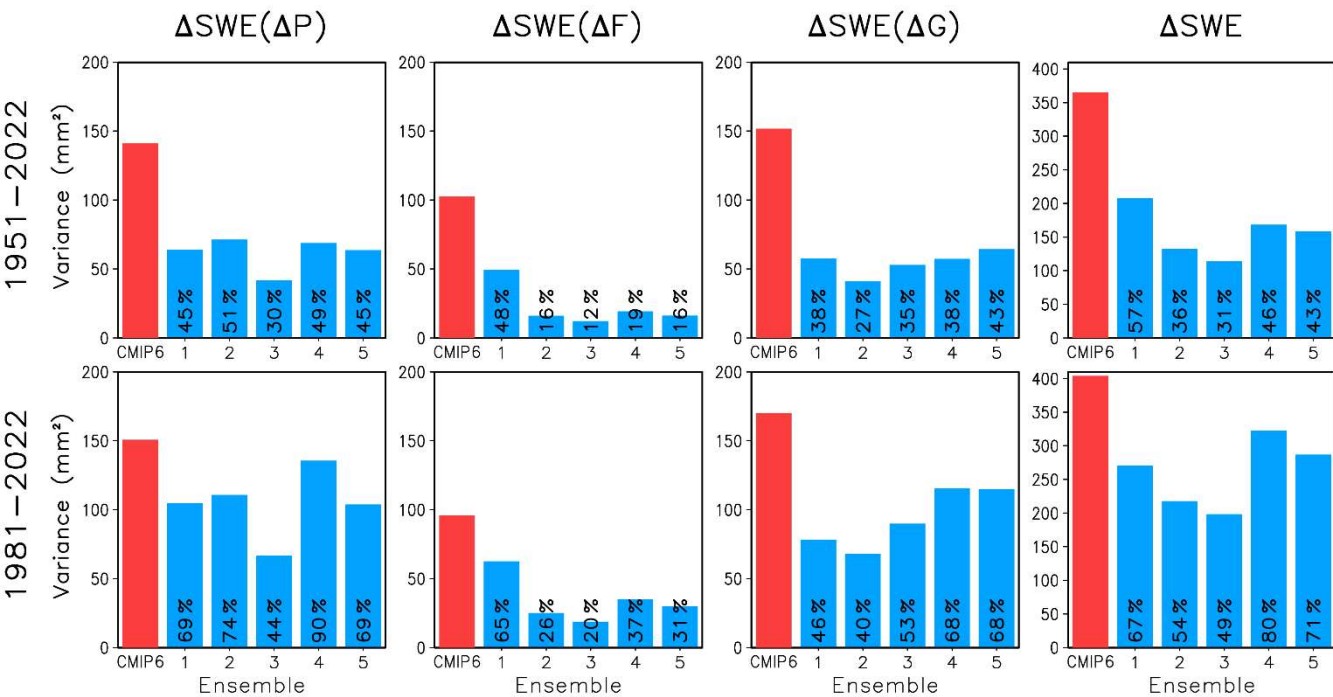

**Figure 5.** Red bars: variance of the SWE trend (column 4) and its three main components (columns 1-3) between the 22 models, as averaged over the Total Snow Area. Blue bars: the variance within each of the five single-model ensembles. The per cent values give the ratio of the single-model variance to the CMIP6 multi-model variance.

The variance within the CMIP6 multi-model ensemble incorporates the effects of model differences and internal variability, whereas the variance in the single-model ensembles only includes the latter. Therefore, the variance in the multi-model

ensemble is larger. Furthermore, the smaller sample size in estimating the trends makes the single-model variances larger in the shorter 1981-2022 than the longer 1951-2022 period. The CMIP6 multi-model variance is more similar between the two periods, because the reduced internal variability is counteracted by larger model-related differences when the length of the period (and hence the change in radiative forcing) increases.

The magnitude of internal variability varies somewhat between the five single-model ensembles. Even so, it seems that internal variability probably explains a majority of the variance in SWE trends in the 22-model CMIP6 ensemble in 1981-2022 (single-model variances 49-80 % of the multi-model variance, bottom-right panel in Fig. 5). Conversely, in 1951-2022, model differences probably dominate (top-right panel in Fig. 5). Similar conclusions hold for the individual trend contributions, except for $\Delta SWE(\Delta F)$ that is much less strongly affected by internal variability than $\Delta SWE(\Delta P)$ and $\Delta SWE(\Delta G)$ in most of the single-model ensembles. This is probably because the snowfall fraction is mainly determined by temperature, and the contribution of internal variability to inter-model differences tends to be smaller for temperature than precipitation changes (Räisänen, 2001; Hawkins and Sutton 2009, 2011; Lehner et al., 2020).

The model-to-reanalysis trend differences are mostly larger than the inter-model differences (Section 7.1). Therefore, internal variability probably explains a smaller fraction of them than of the inter-model differences, assuming that the magnitude of internal variability in the models is realistic. Nonetheless, internal variability is an important complication also when comparing the CMIP6 simulations with the reanalysis data sets.

### 7.3 Changes in winter temperature, precipitation, and atmospheric circulation

To help understand the model-to-reanalysis differences in the trends of SWE and the components of this trend (Eq. 2), as well as the differences between the ERA5L and MERRA2 reanalyses, the trends in November-to-March (NDJFM) mean temperature and precipitation between winters 1981 and 2022 in ERA5L, MERRA2 and the CMIP6 MMM are shown in the first two columns of Fig. 6. The CMIP6 MMM NDFJM mean warming and precipitation increase are both more spatially homogeneous and generally larger in magnitude than the trends in the two reanalyses. The smaller geographic variation is expected, due to the smoothing caused by averaging over the 22 model simulations. However, the trend in the Total Snow Area mean temperature in ERA5L (1.6 °C) is exceeded in 20 of the 22 models and the smaller warming in MERRA2 (0.9 °C) in all 22 models (Table 2). The increase in the Total Snow Area mean precipitation in both reanalyses is also exceeded in all 22 models.

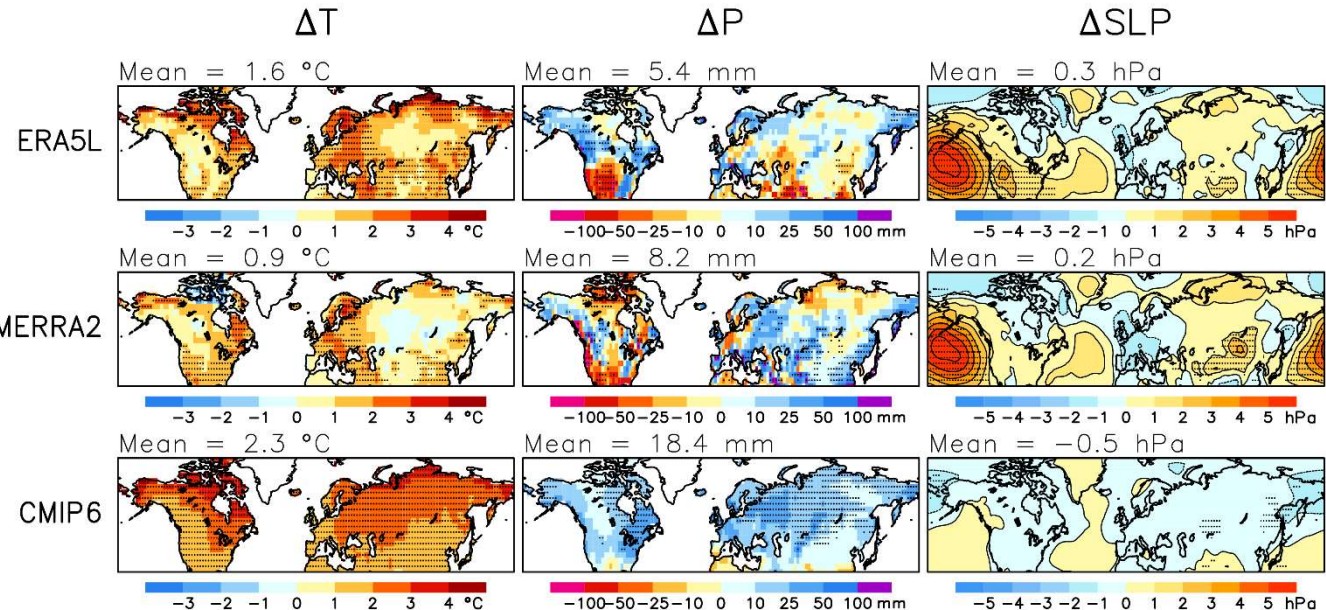

**Figure 6.** Trends in NDJFM mean temperature, precipitation, and sea level pressure between winters 1981 and 2022 in ERA5L (for sea level pressure, ERA5), MERRA2 and the CMIP6 MMM. The area means for the Total Snow Area are given in the headers.

Although the link from temperature and precipitation to SWE is modulated by the seasonally and geographically varying baseline climate (Eq. 2), the trend in NDJFM temperature is a good predictor of the March $\Delta SWE(\Delta F)$ trend within the CMIP6 ensemble, with a correlation of -0.83 for area means in the Total Snow Area. Similarly, the average trends in NDJFM precipitation and $\Delta SWE(\Delta P)$ are strongly correlated ($r = 0.81$). Thus, the overestimation of the positive $\Delta SWE(\Delta P)$ and negative $\Delta SWE(\Delta F)$ trends in the CMIP6 ensemble is consistent with the overestimation of the precipitation and temperature

trends. The tendency for too negative $\Delta SWE(\Delta G)$ trends in the CMIP6 models (column 3 in Fig. 3) also appears physically consistent with the overestimated warming, although the correlation between the area mean temperature and $\Delta SWE(\Delta G)$ trends in the CMIP6 ensemble is weak (-0.19). Similarly, the less negative $\Delta SWE(\Delta F)$ and $\Delta SWE(\Delta G)$ trends in MERRA2 than ERA5L are likely linked to the smaller warming in MERRA2.

To explore whether the CMIP6-to-reanalysis differences might be explained by reanalysis uncertainty, the NDJFM temperature trend was also calculated from the station-based CRU analysis (Fig. B3a) and the precipitation trend from the CRU, GPCC, and GPCP analyses (Figs. B3b-d). The Total Snow Area mean warming in CRU (1.5 °C) is much closer to ERA5L than MERRA2 but substantially below the CMIP6 MMM of 2.3 °C. Similarly, the increases in area mean precipitation in these analyses (9.4 mm in CRU, 6.0 mm in GPCC and 5.2 mm in GPCP) are all well below the CMIP6 average (18.4 mm).

This suggests that the general overestimate of the 1981-to-2022 temperature and precipitation trends in the CMIP6 models is real. However, reanalysis uncertainty might still be an important factor at smaller spatial scales. For example, the spatial

correlation between the CRU and ERA5L trends in the 2.5° grid is 0.69 for temperature but only 0.39 for precipitation (Table B2).

Aside from overestimating the NDJFM climate trends in the Total Snow Area, most of the CMIP6 models also simulate too large global and annual mean warming from August 1980 to July 2022, with the CMIP6 MMM (1.05 °C) exceeding the warming in the ERA5 reanalysis (0.81 °C) by 30 %. Yet the relative difference in the Total Snow Area NDJFM temperature trends is slightly larger (40 %), possibly because of different trends in the atmospheric circulation. The CMIP6 MMM shows a minor decrease in NDJFM mean sea level pressure in most of the Northern Hemisphere continents (bottom-right in Fig. 6).

The pressure trends in ERA5L and MERRA2 agree well with each other and show a more complicated pattern of change, but the trend in most of Eurasia and North America is positive rather than negative. It is tempting to speculate that, in winter when little solar radiation is available, more positive pressure trends in the real world than in the CMIP6 ensemble have acted to reduce cloudiness and increase radiative cooling, thus moderating the warming relative to that in the models. Similarly, both the more positive pressure trends and the smaller warming, which is expected to moderate the increase in atmospheric water

vapour, have likely reduced the precipitation increase relative to that simulated by the models. More quantitative analysis of the circulation-related temperature and precipitation trends would require the use of a dynamical adjustment technique (e.g., Smoliak et al., 2015; Deser et al., 2016; Saffioti et al., 2016; Räisänen, 2021b).

In the longer period from winter 1951 to 2022, the agreement between the ERA5L and the CMIP6 MMM trends is better (Fig.

7). The increase in precipitation is still generally smaller in ERA5L, but the warming is of the same magnitude with the CMIP6 ensemble. The ERA5L pressure trends from winter 1951 to 2022 are quite different from the trends starting in winter 1981 (compare Figs. 7 and 6). Since winter 1951, there has been a widespread pressure decrease in high latitudes and an increase in the subtropical North Atlantic and southern Europe. This pattern is indicative of positive trends in the Arctic and North Atlantic Oscillations, and the resulting enhanced westerly flow from the Atlantic Ocean towards mid-to-high-latitude Eurasia would

be expected to amplify the warming in much of western and central Eurasia (Iles and Hegerl, 2017). Thus, the different pressure trends between the two periods probably partly explain why the average NDJFM warming in the Total Snow Area in ERA5L is closer to the CMIP6 MMM in 1951-2022 than in 1981-2022. Furthermore, the global annual mean warming in CMIP6 agrees better with ERA5 in the August 1950 to July 2022 period (MMM 1.14 °C, ERA5 1.10 °C) than in the August 1980 to July 2022 period (MMM 1.05 °C, ERA5 0.81 °C).

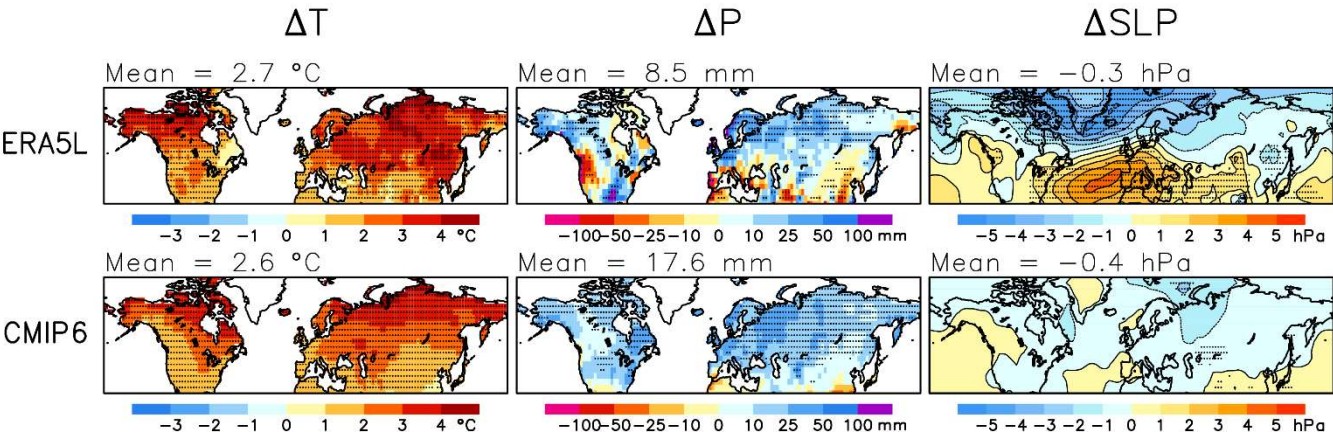

**Figure 7.** Trends in NDJFM mean temperature, precipitation, and sea level pressure between winters 1951 and 2022 in ERA5L (for sea level pressure, ERA5) and the CMIP6 MMM. The area means for the Total Snow Area are given in the headers.

## 8. Conclusions

The SWE at a given time of the winter season depends on the time integral of total precipitation $P$ multiplied by the snowfall fraction $F$ together with the fraction of accumulated snowfall that remains on the ground (snow-on-ground fraction $G$). The present study has applied this framework to diagnose the SWE climates and trends in the ERA5-Land (ERA5L) and MERRA2 reanalyses and 22 CMIP6 climate models, so to reveal their similarities and differences. Comparison with the GlobSnow v3.0 SWE analysis was also included. The focus was on SWE in March, when the Northern Hemisphere snow mass is the largest (Pulliainen et al., 2020). The main findings are summarized below.

**Average SWE climate**. A high degree of similarity was found between ERA5L, MERRA2 and the CMIP6 multi-model mean (MMM), with pairwise spatial correlations of 0.79-0.88 for SWE and 0.88-0.96 for its three multiplicative factors. One noteworthy difference is the much smaller total precipitation and SWE in mountainous areas in MERRA2 than in ERA5L. The GlobSnow SWE estimates agree slightly less well with MERRA2 and ERA5L than these two agree with each other and the CMIP6 MMM. However, although the CMIP6 MMM is within the range of observational uncertainty, there is over a factor of two variation in the area mean SWE between the individual CMIP6 models. The largest contribution to this variation comes from inter-model differences in the snow-on-ground fraction.

**Trends from winter 1951 to 2022.** ERA5L and the CMIP6 models agree qualitatively well on the dynamics of SWE change. Although increasing total precipitation has acted to increase SWE in most of the extratropical Northern Hemisphere (average contribution in ERA5L in the Total Snow Area: 7.1 mm), this has been more than compensated by reduced snowfall fraction (-8.0 mm) and, in most areas, reduced snow-on-ground fraction (-7.4 mm). There is a reasonable spatial correlation (0.51) between the geographical distributions of the March mean SWE trend between ERA5L and the CMIP6 MMM. Both ERA5L

and the CMIP6 models share an increase in SWE in most parts of Alaska, northern Canada, and Siberia, together with decreases

in much of southern Canada, the contiguous United States and Europe, excluding northern Scandinavia in ERA5L (Figure 2).

**Trends from winter 1981 to 2022.** The agreement between ERA5L and the CMIP6 MMM in this period is worse than for the longer period 1951-2022, with a spatial correlation of only 0.09 in the March SWE trend. The correlations between the ERA5L, MERRA2 and (for winters 1981-2018) GlobSnow SWE trends are also modest, suggesting a substantial observational

uncertainty in the SWE change over this period. The spatial patterns of SWE change in all the observational data sets are noisier than those in ERA5L in 1951-2022, which mirrors at least partly a larger contribution from internal variability. However, compared with both ERA5L and MERRA2, the CMIP6 models tend to simulate both larger SWE increases due to increasing precipitation and larger SWE decreases due to decreasing snowfall and snow-on-ground fractions.

**Potential causes of model-to-model and model-to-reanalysis differences in SWE trends and their decomposition.** A substantial fraction of the inter-model variance of the local March mean SWE trends and their decomposition may be caused by internal variability. In the 1981-2022 period this may even exceed the genuinely model-related differences (except for the trend due to changing snowfall fraction), but this is less likely for the longer 1951-2022 trends. Nonetheless, model differences also play a role. Furthermore, particularly in 1981-2022, the trends in ERA5L and MERRA2 tend to differ more from the

CMIP6 trends than the trends in the individual CMIP6 models differ from each other. This suggests a systematic bias either in the models or in the reanalyses. Compared with ERAL and MERRA2, the models generally overestimate the increases in both temperature and precipitation in the extratropical Northern Hemisphere since winter 1981, which qualitatively explains the excessive precipitation-related increases and snowfall-fraction-related decreases of SWE in the CMIP6 ensemble. Both the exaggeration of the recent global mean warming in the models and the differing simulated and observed trends in the

atmospheric circulation since winter 1981 likely contribute to these differences. Alternative observational estimates of temperature and precipitation trends suggest that the general overestimate of winter warming and precipitation increase in CMIP6 is robust to observational uncertainty, although the latter appears more important when considering the trends on smaller spatial scales.

A key Insight from this paper is the relative difficulty of simulating SWE trends correctly in climate models, due to the competition between generally increasing winter precipitation with decreasing snowfall fraction and enhanced snowmelt in a warmer climate. Puzzlingly, the CMIP6 models appear to manage this challenge reasonably well when considering the SWE trends from winter 1951 but less well for the shorter trends since winter 1981. Although probably partly explained by larger internal variability in shorter climate trends, this calls for further research on at least two topics.


1. Considering the differences between the ERA5L, MERRA2 and GlobSnow SWE trends since winter 1981, real-world SWE trends require additional analysis. What causes the differences between these data sets, and what can be concluded on their relative reliability based on in-situ SWE and snow depth observations?

2. How have the different trends in model-simulated and observed atmospheric circulation affected the SWE trends? Several methods of dynamical adjustment have been developed to separate circulation-induced trends of temperature and / or precipitation from underlying thermodynamic changes (e.g., Smoliak et al., 2015; Deser et al., 2016; Saffioti et al., 2016; Räisänen, 2021b). Extending such adjustments to SWE may be more challenging because SWE in (e.g.) March depends on the weather history of the whole winter season, rather than on the weather in March alone. One approach might be to first construct modified time series of temperature, precipitation, and other necessary weather parameters, from which the effects of circulation variability are eliminated using dynamical adjustment. Then, these modified time series could be used to drive land surface models such as those used in ERA5L and MERRA2, so to simulate the potential evolution of SWE in the absence of circulation trends.

Another important question is, how water resource managers and other stakeholders needing SWE projections should use the information available from climate model ensembles. On the face of it, the answer seems disappointing. As real-world SWE trends appear to have hitherto been further away from the model-simulated trends than the latter are from each other, this might also well apply to the trends in the future. It would therefore seem prudent to allow for all the uncertainty suggested by the variation between climate models, possibly adding a safety margin for systematic model errors. However, this conclusion may need re-evaluation if future research is able to reduce the uncertainty in the observed trends. Moreover, the practice of giving all models the same weight in projections of future SWE change is probably sub-optimal, particularly for longer-term projections in which model differences grow increasingly dominant over internal variability. For example, Räisänen (2008) found a dependence between model-simulated baseline winter temperatures and projected future SWE changes, which makes increases of SWE more likely in models with a cold than a warm temperature bias. He also used inter-model cross validation to show that this physically expected dependence could be potentially used for improving probabilistic projections of SWE change. Furthermore, inter-model variations in the Northern Hemisphere snow albedo feedback are strongly correlated between seasonal and climate change time scales (Hall and Qu, 2006; Qu and Hall, 2014). This does not guarantee that a similar cross-time-scale constraint would be available for SWE as well, but it suggests that the possibility is worth exploring.

**Appendix A. Further details on the observational data sets**

The Uniform Resource Locator (URL) addresses from which the various observational data sets were downloaded are listed in Table A1. Additional notes are given in the text that follows.

**Table A1.** URL addresses, references and dates of last access for the observational data sets

| Data set acronym | URL and reference | Last access |
|---|---|---|
| ERA5L | https://cds.climate.copernicus.eu/cdsapp#!/dataset/reanalysis-era5-land-monthly-means (C3S, 2023a) | 8 Mar 2023 |
| MERRA2 | https://disc.gsfc.nasa.gov/datasets?project=MERRA-2 (GES DISC, 2023) | 8 Mar 2023 |
| CRU | https://crudata.uea.ac.uk/cru/data/hrg/cru_ts_4.06/cruts.2205201912.v4.06/ (CRU, 2023) | 8 Mar 2023 |
| GPCC | https://opendata.dwd.de/climate_environment/GPCC/full_data_monthly_v2022/05/ (DWD, 2023) | 8 Mar 2023 |
| GPCP | https://www.ncei.noaa.gov/data/global-precipitation-climatology-project-gpcp-monthly/access/ (NCEI, 2023) | 8 Mar 2023 |
| GlobSnow | https://doi.pangaea.de/10.1594/PANGAEA.911944 (Luojus et al., 2020) | 8 Mar 2023 |
| ERA5 | https://cds.climate.copernicus.eu/cdsapp#!/dataset/reanalysis-era5-single-levels-monthly-means (C3S, 2023b) | 8 Mar 2023 |

**A1 Regridding GlobSnow to the 2.5° × 2.5° latitude-longitude grid**

The GlobSnow SWE data were first regridded from their original 25 km equal area grid to a 0.25° × 0.25° latitude-longitude grid using the nearest-neighbour method. The values in the 0.25° × 0.25° grid were then averaged to 2.5° × 2.5° latitude-longitude boxes, excluding those (sea or mountainous) 0.25° × 0.25° grid boxes in which the data were missing. If missing data covered more than half of a 2.5° × 2.5° grid box, the value for that 2.5° × 2.5° grid box was left undefined.

**A2 Extension of CRU and GPCC precipitation time series until July 2022**

At the time of writing, the CRU precipitation data were available until December 2021 and the GPCC data until December 2020. To allow the estimation of trends until the snow year 2021/22, these time series were extended to July 2022 by calculating, at each grid box and month separately, the ratio between the CRU (GPCC) and ERA5L mean precipitation in the years 2011-2021 (2011-2020) and multiplying the ERA5L precipitation for the remaining 7 or 19 months by this ratio.

   **Appendix B. Additional results**

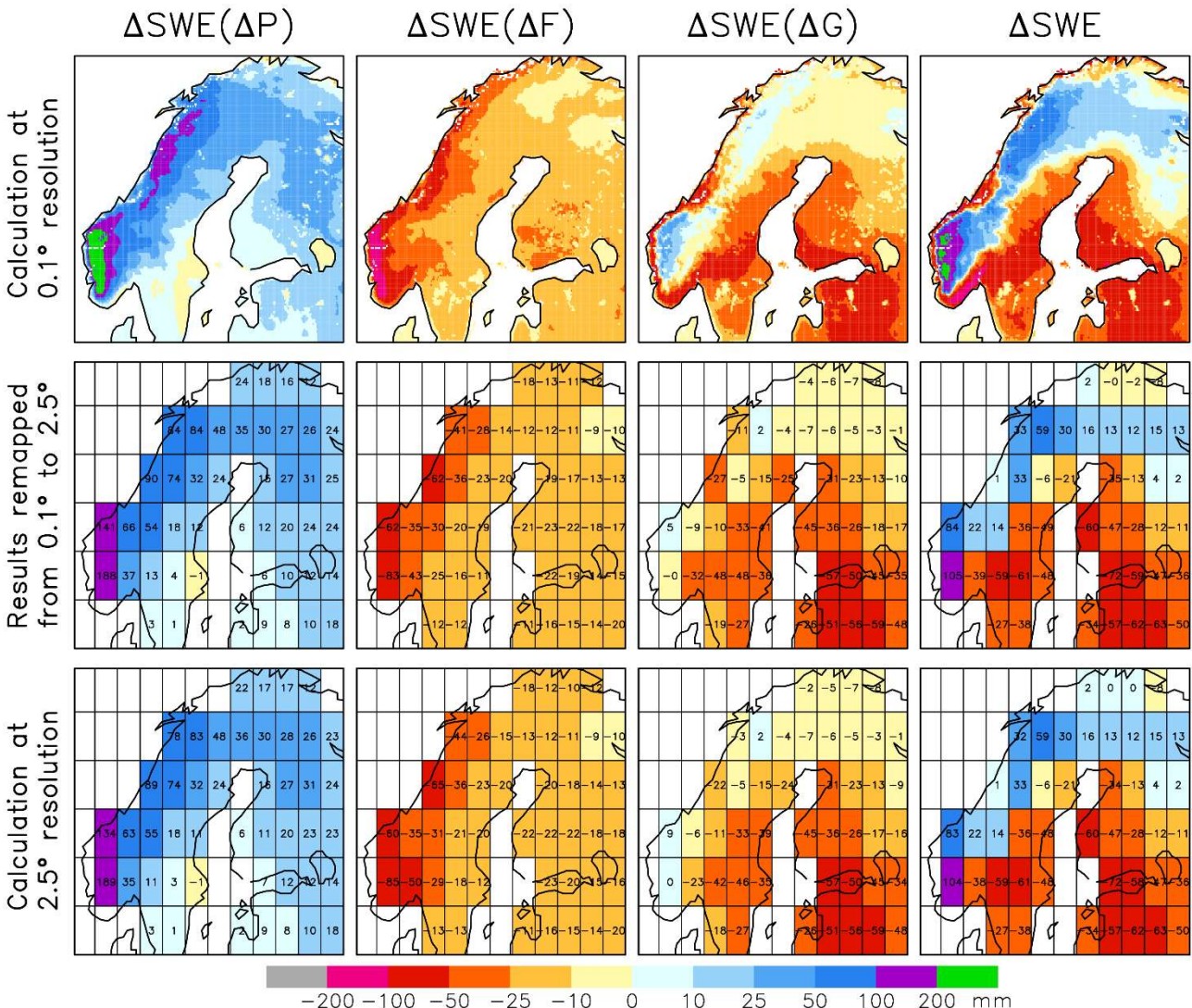

**Figure B1**. Trend in March mean SWE in Scandinavia from 1951 to 2022 (mm (71 yr)$^{-1}$) (column 4) and the contributions to it from changes in total precipitation (column 1), snowfall fraction (column 2) and snow-on-ground fraction (column 3). Top: calculation using ERA5L data at their native 0.1° resolution. Middle: values from the top row remapped to 2.5° resolution. 655   Bottom: calculation using ERA5L data remapped to 2.5° resolution before the trend decomposition.

**Table B1.** Average March mean SWE trends (mm) in Eurasia and North America. The values without parentheses represent trends until 2022 in the Total Snow Area and those in parentheses trends until 2018 in the GlobSnow Area.

|  | Trend from 1951 to 2022 | | Trend from 1981 to 2022 (2018) | |
|---|---|---|---|---|
|  | Eurasia | North America | Eurasia | North America |
| ERA5L | -7.6 | -10.0 | -12.6 (-12.8) | 4.7 (1.2) |
| MERRA2 |  |  | -1.3 (-4.5) | -6.1 (-7.1) |
| CMIP6 MMM | -4.1 | -10.8 | -3.7 (-2.4) | -9.6 (-6.5) |
| GlobSnow |  |  | (0.5) | (-16.9) |

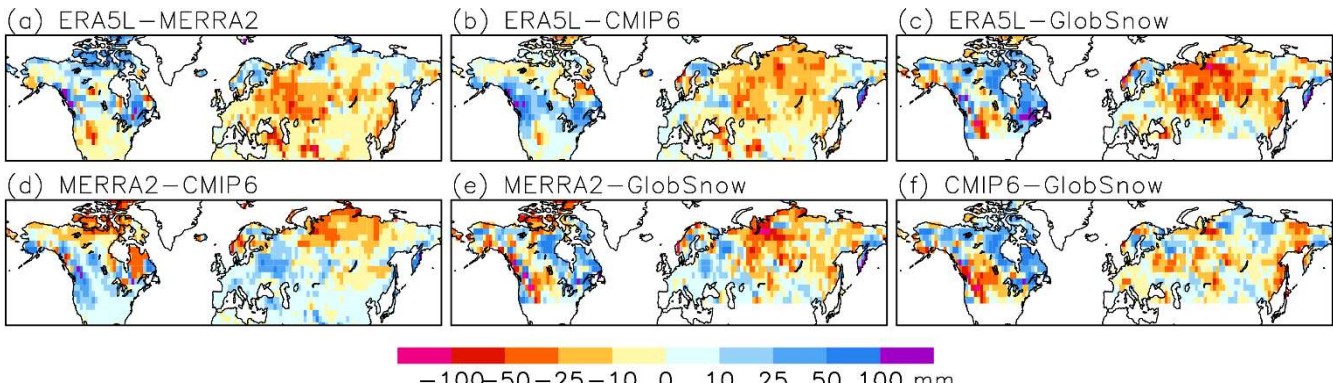

**Figure B2.** Differences in March mean SWE trends between the data sets identified in the map headers. Periods and units: (a, b, d) 1981 to 2022 (mm (41 yr)$^{-1}$); (c, e, f) 1981 to 2018 (mm (37 yr)$^{-1}$).

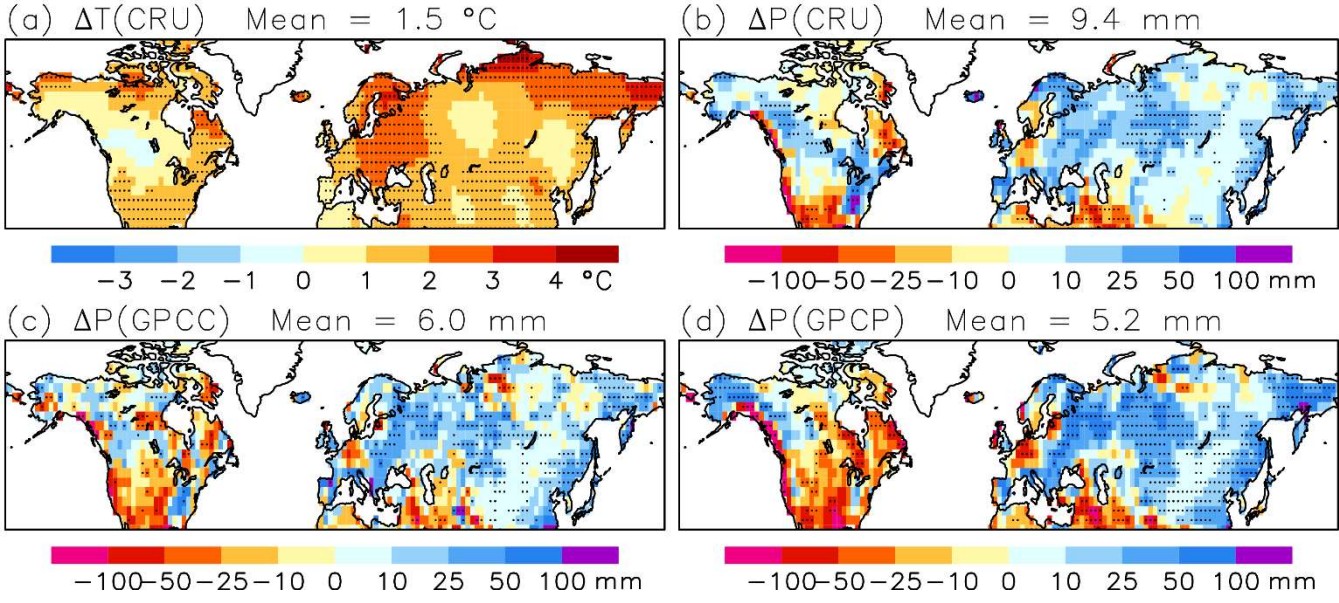

**Figure B3.** Trends in NDJFM mean climate between winters 1981 and 2022. (a) Temperature in the CRU analysis, and (b)-(d) precipitation in the (b) CRU, (c) GPCC, and (d) GPCP analyses. The stippling indicates areas where the trends are statistically significant against interannual variability (90% level, two-sided $t$ test). The area means for the Total Snow Area are given in the headers.

**Table B2.** Total Snow Area spatial correlation of NDJFM mean temperature and precipitation trends from winter 1981 to 2022 between different data sets.

| Temperature | MERRA2 | CMIP6 | CRU | | |
|---|---|---|---|---|---|
| ERA5L | 0.33 | 0.60 | 0.69 | | |
| MERRA2 | | -0.04 | 0.33 | | |
| CMIP6 MMM | | | 0.48 | | |
| Precipitation | MERRA2 | CMIP6 | CRU | GPCC | GPCP |
| ERA5L | 0.28 | 0.28 | 0.39 | 0.41 | 0.21 |
| MERRA2 | | 0.00 | 0.41 | 0.46 | 0.41 |
| CMIP6 MMM | | | 0.11 | 0.01 | -0.07 |
| CRU | | | | 0.65 | 0.67 |
| GPCC | | | | | 0.70 |

**Code and data availability.** All observational data sets used in this study are publicly available, as detailed in Table A1. The CMIP6 simulations are available from the Earth System Grid Federation (https://esgf-node.llnl.gov/search/cmip6/, ESGF, 2023). The post-processed data and GrADS (Grid Analysis and Display System) scripts needed for reproducing the figures and numerical results in this article are available at https://doi.org/10.5281/zenodo.7707302 (Räisänen, 2023).

**Author contribution.** JR did all the work for this study.

**Competing interests.** The author declares that he has no conflict of interest.

**Acknowledgments**. The author thanks Kari Luojus for discussions on the GlobSnow data set and the three anonymous reviewers for their constructive comments.

**Financial Support.** This research was supported by Academy of Finland Flagship funding (grant no. 337549). Open-access funding was provided by the Helsinki University Library.

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
