# Peer review of "Changes in March mean snow water equivalent since the midtwentieth century and the contributing factors in reanalyses and CMIP6 climate models"

_The Cryosphere, 2022_

## Author Comment (AC1)

**Response to Reviewer 1**

I thank the reviewer for his / her valuable comments on the manuscript. My response to the comments and the changes to be made in the revised manuscript are detailed below. For clarity, the comments are in blue font, while my response is in black. In some cases, I have included the text planned to appear in the revised manuscript in red font.

General comments:

This interesting research documents the impact of different factors on changes in March SWE by analyzing two reanalyses products and 22 CMIP6 models. The study shows that SWE is decreasing over most of the Northern Hemisphere, as decreases in snowfall and snow-on-ground are more significant than the increase in total precipitation. However, there are large variations between the products analyzed. In general, the study is wellwritten and easy to follow, and I only have a few comments as outlined below.

As the author discusses, there are large differences between different models/reanalyses analyzed in this research and making definitive conclusions might be difficult but, as the paper discusses changes in SWE, I think it might be good to have a bit more clear statements about these changes in the conclusion.

The following characterization of the geographical distribution of the 1951-to-2022 SWE trends will appear in the Conclusions section of the revised manuscript. Repeating this for the more noisy 1981-to-2022 trends would become too complicated.

**Trends from winter 1951 to 2022**. … Both ERA5L and the CMIP6 models share an increase in SWE in most parts of Alaska, northern Canada, and Siberia, together with decreases in much of southern Canada, the contiguous United States and Europe, excluding northern Scandinavia in ERA5L (Figure 2).

Also, some comments on how the changes in March mean SWE differ from the findings of other papers mentioned in the introduction (e.g., Pulliainen et al. (2020) and Mudryk et al. (2020)) might be good. This research has some similarities to Kouki et al. (2022) which analyses SWE in CIMP6 models and the effects of different factors (temperature and precipitation), though this article covers a larger time span and includes also analyses of reanalysis products. This manuscript mentions that CMIP6 SWE exceeds the GlobSnow estimates similarly to Kouki et al. (2022) but maybe a few more words about any similarities/differences would be good...?

The last two paragraphs in Section 6 of the revised manuscript will include some comparison of the SWE trends to Mudryk et al. (2020) and Pulliainen et al. (2020). Following one of your later comments, the second of these paragraphs makes this comparison separately for Eurasia and North America. **The revised Table 5, the new Table B1, and the new Figure B2, which the text refers to, are included in the end of this file**.

The CMIP6 MMM Total Snow area mean SWE trend in March 1981-2022 (-5.8 mm) falls between the trends in ERAL (-6.9 mm) and MERRA2 (-3.1 mm), whereas the corresponding trend in March 1981-2018 in the GlobSnow area (-3.8 mm) is slightly less negative than those in ERA5L, MERRA2 and GlobSnow (-5.4 to -8 mm). The Mudryk et al. (2020) consensus estimate of Northern Hemisphere March mean SWE trend in the same period suggests an even larger decrease (~9 mm by unit conversion from their Fig. 1c).

The SWE trends in the various data sets have higher spatial correlations in Eurasia than in North America (last two columns of Table 5). However, there is a striking discrepancy in both the Eurasian and North American

area mean 1981-to-2022 and 1981-to-2018 SWE trends between ERA5L and the other data sets (Table B1). ERA5L suggests an increase in average SWE in North America and a major decrease in Eurasia, while MERRA2, GlobSnow and CMIP6 MMM all indicate larger decreases in North America than Eurasia. In particular, GlobSnow shows a near-zero SWE trend in Eurasia but a 17 mm decrease from 1981 to 2018 in North America, in good agreement with Pulliainen et al. (2020). Mirroring these mean values, maps of the inter-data-set trend differences (Fig. B2) reveal a particularly pervasive difference between ERA5L and GlobSnow, with more negative trends in ERA5L in much of Eurasia but more positive trends in North America (Fig. B2c). Nonetheless, in the longer 1951-to-2022 period, the SWE decrease in ERA5L is also slightly larger in North America than in Eurasia (Table B1).

Regarding comparison to Kouki et al. (2022), the following text will be added to Section 4.

The CMIP6 22-model mean SWE in the Total Snow Area is close to MERRA2 but 15 % below ERA5L; in the GlobSnow Area it is also below MERRA2. The average precipitation in the CMIP6 models exceeds both ERA5L and MERRA2, but this is compensated by lower mean values of F* and G (third row in Fig. 1). On the other hand, the average CMIP6 SWE exceeds the GlobSnow estimate (bottom-right corner in Fig. 1) by nearly 10 %. Kouki et al. (2022) also used GlobSnow as their main observational data set, finding an average ~15 % overestimate of March mean SWE for a larger set of 38 CMIP6 models (their Fig. 2b). Both the different sets of models and the inclusion of mountainous areas (where GlobSnow was replaced by other observational estimates) by Kouki et al. (2022) may contribute to this slight difference. Using linear regression, Kouki et al. (2022) attributed the overestimate of simulated SWE in February to too large November-to-January precipitation in the CMIP6 models. Although they made this regression analysis for February rather than March mean SWE, this result is in line with the CMIP6 MMM overestimate of area mean P* suggested by Fig. 1.

Specific comments:

L11: 'This is repeated…' This sentence is a bit unclear, what is repeated?

This will be clarified as: This trend attribution is repeated

L130: A sentence or two describing MERRA 2 might be good (as there is a short description of ERA5-Land).

This sentence will be added to Section 2: MERRA2 is an atmosphere-land reanalysis produced by version 5.12.4 of the Goddard Earth Observing System atmospheric data assimilation system.

Table 1.: Possibly line numbers have jumped to the last column of the table? Or what are the numbers (145, 150) in the table?

Yes, they are line numbers. Unfortunately, as this jump was generated in the conversion from the MS Word document to a pdf file, I can do little to prevent its possible re-appearance in the revised manuscript.

Figure 3: It might be interesting to see the difference in SWE between different versions? Different spatial trends are visible in the figures but the same figures showing the differences might be beneficial.

These differences are shown in the new **Figure B2, included in the end of this file**.

L520: Are there some areas where the spatial correlation is better? The average SWE values are larger in North America and some products have problems with this and thus have better performance in Eurasia.

I made the comparison separately for Eurasia and North America. In line with your speculation, the spatial correlations between the different data sets are better in Eurasia. However, there is a major discrepancy between the continental mean SWE trends between ERA5-Land and the other data sets. This will be discussed in the last paragraph of Section 6 in the revised manuscript. The revised Table 5, the new Table B1 and the new Figure B2 are in the end of this file.

The SWE trends in the various data sets have higher spatial correlations in Eurasia than in North America (last two columns of Table 5). However, there is a striking discrepancy in both the Eurasian and North American area mean 1981-to-2022 and 1981-to-2018 SWE trends between ERA5L and the other data sets (Table B1). ERA5L suggests an increase in average SWE in North America and a major decrease in Eurasia, while MERRA2, GlobSnow and CMIP6 MMM all indicate larger decreases in North America than Eurasia. In particular, GlobSnow shows a near-zero SWE trend in Eurasia but a 17 mm decrease from 1981 to 2018 in North America, in good agreement with Pulliainen et al. (2020). Mirroring these mean values, maps of the inter-data-set trend differences (Fig. B2) reveal a particularly pervasive difference between ERA5L and GlobSnow, with more negative trends in ERA5L in much of Eurasia but more positive trends in North America (Fig. B2c). Nonetheless, in the longer 1951-to-2022 period, the SWE decrease in ERA5L is also slightly larger in North America than in Eurasia (Table B1).

L521-523: "However, compared with both ERA5L and MERRA2…" I find this sentence bit unclear, what does overestimating the SWE decrease mean?

To be reformulated as:

However, compared with both ERA5L and MERRA2, the CMIP6 models tend to simulate both larger SWE increases due to increasing precipitation and larger SWE decreases due to decreasing snowfall and snow-on-ground fractions.

**Revised Table 5** (with two new columns on the right):

**Table 5.** Spatial correlation of the trend in March mean SWE and its contributing terms (Eq. 2) between different data sets. SWE-EUR and SWE-NAM refer to SWE trends in Eurasia and North America, respectively. The values without (within) parentheses represent the Total Snow Area (GlobSnow Area).

| Years | | $\Delta SWE(\Delta P)$ | $\Delta SWE(\Delta F)$ | $\Delta SWE(\Delta G)$ | SWE | SWE-EUR | SWE-NAM |
|---|---|---|---|---|---|---|---|
| 1951-2022 | ERA5L vs. CMIP6 | 0.45 | 0.75 | 0.58 | 0.51 | 0.59 | 0.49 |
| 1981-2022 | ERA5L vs. MERRA2 | 0.42 | 0.79 | 0.39 | 0.29 (0.48) | 0.52 (0.70) | 0.11 (0.15) |
| | ERA5L vs. CMIP6 | 0.35 | 0.61 | 0.27 | 0.09 (0.16) | 0.45 (0.37) | -0.34 (-0.12) |
| | MERRA2 vs. CMIP6 | 0.17 | 0.57 | 0.30 | 0.16 (0.12) | 0.23 (0.21) | 0.02 (-0.05) |
| 1981-2018 | ERA5L vs. GlobSnow | | | | (0.13) | (0.41) | (0.02) |
| | MERRA2 vs. GlobSnow | | | | (0.24) | (0.31) | (0.16) |
| | CMIP6 vs. GlobSnow | | | | (0.34) | (0.51) | (0.07) |

**New Table B1**:

Table B1. Average March mean SWE trends (mm) separately in Eurasia and North America. The values without parentheses represent trends until 2022 in the Total Snow Area and those in parentheses trends until 2018 in the GlobSnow Area. CMIP6 = CMIP6 MMM.

|  | Trend from 1951 to 2022 | | Trend from 1981 to 2022 (2018) | |
|---|---|---|---|---|
|  | Eurasia | North America | Eurasia | North America |
| ERA5L | -7.6 | -10.0 | -12.6 (-12.8) | 4.7 (1.2) |
| MERRA2 |  |  | -1.3 (-4.5) | -6.1 (-7.1) |
| CMIP6 | -4.1 | -10.8 | -3.7 (-2.4) | -9.6 (-6.5) |
| GlobSnow |  |  | (0.5) | (-16.9) |

**New Figure B2**:

[Figure]

Figure B2. Differences in March mean SWE trends between the data sets identified in the map headers. Periods and units: (a, b, d) 1981 to 2022 (mm (41 yr)$^{-1}$); (c, e, f) 1981 to 2018 (mm (37 yr)$^{-1}$).

---

## Author Comment (AC2)

**Response to Reviewer 2**

I thank the reviewer for his / her valuable comments on the manuscript. My response to the comments and the changes to be made in the revised manuscript are detailed below. For clarity, the comments are in blue font, while my response is in black. In some cases, I have included the text planned to appear in the revised manuscript in red font.

The author compares how three contributing factors that influence March historical snow water equivalent trends are represented in two reanalysis products and in 22 CMIP6 models. The factors contributing to total SWE change are changes related to the total precipitation, the fraction of precipitation occurring as snowfall, and the fraction of accumulated snowfall which remains on the ground. The author determines that there is broad agreement in the subcomponents over the long analysis period assessed (1951-2022) but that internal variability and model biases reduce the agreement between historical trend estimates and the CMIP6 multi-model mean over the shorter 1981-2022 period analyzed.

General Comments: Overall this paper is well-written and provides logically argued results. The framework used to attribute SWE changes to changes in precip, snowfall fraction, and snow-on-ground seems to works well and provide reasonable results. I only have a few issues I'd like to see addressed or commented on.

Specific comments:

The snow-on-ground term G absorbs a wide range of processes that will vary substantially from model to model. While inter-model difference in snow melt will be a key component of this term, differences in sublimation and any other structural differences in how a particular model treats snow will also affect it (e.g. vegetation-snow interactions, differences in how well snow water mass is locally conserved in a given model). Thus, it isn't surprising that the models disagree the most on the changes in this term. I think it's worth pointing this out in the text when you first discuss the equations and also at line 270 where you attribute the differences in G primarily to snow melt.

This is an excellent comment. The snow-on-ground fraction is indeed affected by a large number of factors. In response to this comment and related comments from Reviewer 3, the following paragraph will be added to the end of Section 4 in the revised manuscript:

Table 4 also shows that the inter-model differences in G are in relative terms larger than those in P* and F*. This is perhaps unsurprising, since G may be affected by a multitude of factors. As defined by Eq. (1), G reflects the balance between the source (accumulated snowfall) and sinks (snowmelt plus sublimation) of snow. The accumulated snowfall depends on both the amount and phase of precipitation, whereas snowmelt and sublimation are ultimately determined by the amount of energy that the land surface model allocates to them. The latter, in turn, depends on the radiative energy input from the atmosphere, the exchange of sensible and latent heat between the land surface and the atmospheric models, the description of the surface albedo and emissivity, and the use or release of energy associated with temperature changes within the snow-ground-vegetation system. As many of these processes are described differently in different land surface models, it is perhaps unsurprising that the simulated SWE may vary substantially even between land surface models that share the same atmospheric forcing (Mudryk et al., 2015). A more detailed understanding of the causes of variation of G within the CMIP6 ensemble is an important target for future research.

Line 186: I believe you are defining X1=X_clim and X2 = X_clim + dx where dX is the monthly anomaly. Using these definitions x-bar is the monthly mean value plus half the monthly anomaly. If you actually used G_bar = G_clim + 0.5dG, F_bar = F_clim + 0.5dF, P_bar+0.5dP in equation (2) please state this more explicitly.

Yes, this is exactly what was done. I tried to clarify this with the last, bolded sentence in the refined description of Equation (2):

The monthly mean values of $X$ = $SWE$, $G$, $F$ and $P$ over the whole analysis period are denoted as $X_1$, whereas their values in an individual winter are denoted as $X_2$. By further defining $\bar{X} = (X_1 + X_2)/2$ and $\Delta X = X_2 - X_1$, one obtains

$$\Delta SWE(t) = \underbrace{\bar{G} \int_{t_0}^{t} \bar{F} \Delta P dt'}_{\Delta SWE(\Delta P)} + \underbrace{\bar{G} \int_{t_0}^{t} \Delta F \bar{P} dt'}_{\Delta SWE(\Delta F)} + \underbrace{\Delta G \int_{t_0}^{t} \bar{F} \bar{P} dt}_{\Delta SWE(\Delta G)} + \underbrace{\tfrac{1}{4} \Delta G \int_{t_0}^{t} \Delta F \Delta P dt'}_{\Delta SWE(NL)} \qquad (2)$$

Thus, the anomaly in SWE is decomposed to contributions from the total precipitation ($\Delta P$), snowfall fraction ($\Delta F$) and snow-on-ground fraction anomalies ($\Delta G$), plus a non-linear term that is typically two orders of magnitude smaller than the others. **However, there is an implicit non-linearity in the coefficients $\bar{G}$, $\bar{F}$ and $\bar{P}$ in (2) since, for example, $\bar{G} = G_1 + \Delta G/2 \neq G_1$.**

No, I did not do it this way.

I think the MMM trends you show are probably a good representation of the ensemble average given the range of models. But it would still be good to check that the figures look similar if you plot the median trend value at each location in case there is any undue influence in the mean from outlier trends.

I followed this suggestion. The trend maps are very similar; for example, the Total Snow Area spatial correlation between the CMIP6 mean and median trends in SWE is 0.97 in both the 1951-to-2022 and 1981-to-2022 periods. I chose to not include this analysis in the manuscript but will add the CMIP6 median values to Table 4 about the area mean statistics. These are indeed very similar with the multi-model mean values.

L541: Please rephrase. I don't think snow melt occurs more efficiently as the climate warms. There's just increased snowmelt in a warmer climate.

"Increased snowmelt" is also a slightly problematic formulation because, as integrated over the whole winter, the total amount of snowmelt is the same as the accumulated snowfall (which may either increase or decrease). Therefore, this will be reformulated as "enhanced snowmelt in a warmer climate".

---

## Author Comment (AC3)

**Response to Reviewer 3**

is within the scope of The Cryosphere.

5

I thank the reviewer for his / her valuable comments on the manuscript. My response to the comments and the changes to be made in the revised manuscript are detailed below. For clarity, the comments are in blue font, while my response is in black. In some cases, I have included the text planned to appear in the revised manuscript in red font.

This is one of the most thoughtful and comprehensive reviews that I have ever received for any of my manuscripts. Due to practical constraints, I unfortunately cannot implement the two major requests of the
reviewer (regarding spatial and temporal resolution and the more detailed interpretation of the snow-on-ground-fraction term) to the extent that the reviewer might have hoped. However, I have done my best to do what can reasonably be done, and I hope that my justifications for not doing more make good sense.

The manuscript "Changes in March mean snow water equivalent since the mid-twentieth century and the
contributing factors in reanalyses and CMIP6 climate models" uses a decomposition-based approach to evaluate and diagnose changes in historic snow water equivalent (SWE) trends in the northern hemisphere for two historic periods in two reanalysis products, the GlobSnow product, and the CMIP6 ensemble. SWE is decomposed into contributions from total precipitation, snow fraction, and snow-on-ground fractions. The primary findings include high degrees of spatial correlations but individual variation due to model-to-model
differences in snow-on-ground fraction. Total SWE has gone up in many far northern locations due to increases in precipitation, though CMIP6 appears to overestimate SWE increases. Biases in all data sources are discussed, and the paper highlights how difficult SWE is to correctly simulate in climate models. The topic

Overall, I like the methodology as an interesting approach to breaking down drivers of SWE. However, my first major concern centers around choices in the spatial aggregation step and the monthly time step that may lead to very different results. Huge efforts and computational resources have been expended on improving spatial resolution of snow (and many key climate system variables) in global models, and it is unfortunate to not see how finer resolution may improve the insight provided by the work (it also may not, but it needs to be shown). My second major concern is the lack of energy balance components in SWE trends

is perhaps implicit in the G term, but this is not apparent from the methodology description.

My response to these comments is divided to three parts, labeled as (1)-(3) below.

35 (1) I start from the G term because this explanation may also shed light on why the time resolution is not important. In this study, G is a purely diagnostic quantity. From Eq. (1) one gets

$$G = \frac{SWE}{\int_{t_0}^{t} FPdt'} = \frac{SWE}{\int_{t_0}^{t} Snowfall \, dt'} = \frac{SWE}{accumulated \, snowfall}$$

40 For calculating *G*, one only needs the SWE and the accumulated (i.e., time-integrated) snowfall from the beginning of the winter season. Both the SWE and the snowfall are directly available from reanalysis and climate model output. To clarify, the paragraph preceding Eq. (1) will be rewritten as:

Our diagnostic framework follows Räisänen (2008, 2021a). The only three variables that are needed from a
 reanalysis or a model simulation are monthly means of SWE, snowfall, and total precipitation (*P*). The monthly snowfall is first rewritten as *FP*, where *F* is the fraction of precipitation that falls as snow. SWE in month *t* then becomes

**$SWE = G \int_{t_0}^t FPdt'$**

- 50 Clearly, there are many different processes and meteorological factors that affect G. A detailed analysis of these factors in the individual reanalyses and climate models is well beyond of what can be reasonably included in the current paper. However, I will point out the importance of this issue by adding the following paragraph to the end of Section 4 in the revised manuscript:
- 55 Table 4 also shows that the inter-model differences in G are in relative terms larger than those in P\* and F\*. This is perhaps unsurprising, since G may be affected by a multitude of factors. As defined by Eq. (1), G reflects the balance between the source (accumulated snowfall) and sinks (snowmelt plus sublimation) of snow. The accumulated snowfall depends on both the amount and phase of precipitation, whereas snowmelt and sublimation are ultimately determined by the amount of energy that the land surface model allocates to
- 60 them. The latter, in turn, is constrained by the downward solar and thermal radiation reaching the surface, the exchange of sensible and latent heat between the land surface and the atmospheric models, the description of the surface albedo and emissivity, and the use or release of energy associated with temperature changes within the snow-ground-vegetation system. As many of these processes are described differently in different land surface models, it is perhaps unsurprising that the simulated SWE may vary
- 65 substantially even between land surface models that share the same atmospheric forcing (Mudryk et al., 2015). A more detailed understanding of the causes of variation of *G* within the CMIP6 ensemble is an important target for future research.

(2) The low spatial resolution of the analysis leads to a loss of potentially valuable local information from the
 high-resolution analysis products. However, the 2.5° × 2.5° grid is sufficient when the focus is on large-scale trends, as is the case in this paper. Importantly, the trend results in this coarser grid are insensitive to the order between grid remapping and the decomposition (Eqs. (1)-(2) in the manuscript). In other words, if one starts with remapping all the required input data to the 2.5° grid before applying Eqs. (1)-(2), the resulting trends are very similar to those obtained by first doing the calculations in a finer grid and then remapping the trends to the 2.5° grid. The only (expected) exception are 2.5° grid boxes that are partly covered by sea

75 the trends to the 2.5° grid. The only (expected) exception are 2.5° grid boxes that are partly covered by sea in the original grid, because the required input data are only available over land. This similarity, as well as the loss of small-scale information, is illustrated in the **new Figure B1, included in the next page**.

In any case, redoing the analysis on a finer resolution would have a low benefit-to-effort ratio, because I have
 only archived a large fraction of my CMIP6 data in the 2.5° grid. Furthermore, as most of the CMIP6 models
 only have a resolution of 1-2°, using a grid finer than 2.5° would not help to draw much new information
 from the models.

---

## Author Response (AR1)

**Response to Reviewers**

I thank the three reviewers for their constructive comments on my manuscript. My response follows below.

For clarity, I repeat the reviewer comments in **blue** font, whereas my response is in **black**. The changes made to the manuscript are indicated in **red**. When referring to line numbers in the revised manuscript, the line numbers in the no-tracked-changes version are given first and the corresponding line numbers in the tracked-changes version after that. For example, L573-575 / L625-627 means lines 573-575 in the no-tracked-changes version and lines 625-627 in the tracked-changes version.

**Response to Reviewer 1**

General comments:

This interesting research documents the impact of different factors on changes in March SWE by analyzing two reanalyses products and 22 CMIP6 models. The study shows that SWE is decreasing over most of the Northern Hemisphere, as decreases in snowfall and snow-on-ground are more significant than the increase in total precipitation. However, there are large variations between the products analyzed. In general, the study is wellwritten and easy to follow, and I only have a few comments as outlined below.

As the author discusses, there are large differences between different models/reanalyses analyzed in this research and making definitive conclusions might be difficult but, as the paper discusses changes in SWE, I think it might be good to have a bit more clear statements about these changes in the conclusion.

**Response**: the following characterization of the geographical distribution of the 1951-to-2022 SWE trends has been added to the Conclusions section of the revised manuscript. Repeating this for the more noisy 1981-to-2022 trends would become too complicated.

L573-575 / L625-627: Both ERA5L and the CMIP6 models share an increase in SWE in most parts of Alaska, northern Canada, and Siberia, together with decreases in much of southern Canada, the contiguous United States and Europe, excluding northern Scandinavia in ERA5L (Figure 2).

Also, some comments on how the changes in March mean SWE differ from the findings of other papers mentioned in the introduction (e.g., Pulliainen et al. (2020) and Mudryk et al. (2020)) might be good. This research has some similarities to Kouki et al. (2022) which analyses SWE in CIMP6 models and the effects of different factors (temperature and precipitation), though this article covers a larger time span and includes also analyses of reanalysis products. This manuscript mentions that CMIP6 SWE exceeds the GlobSnow estimates similarly to Kouki et al. (2022) but maybe a few more words about any similarities/differences would be good...?

The last two paragraphs in Section 6 of the revised manuscript includes some comparison of the SWE trends to Mudryk et al. (2020) and Pulliainen et al. (2020). Following one of your later comments, the second of these paragraphs makes this comparison separately for Eurasia and North America.

L394-408 / L444-458. The CMIP6 MMM Total Snow area mean SWE trend in March 1981-2022 (-5.8 mm) falls between the trends in ERAL (-6.9 mm) and MERRA2 (-3.1 mm), whereas the corresponding trend in March

1981-2018 in the GlobSnow area (-3.8 mm) is slightly less negative than those in ERA5L, MERRA2, and GlobSnow (-5.4 to -8 mm). The Mudryk et al. (2020) consensus estimate of Northern Hemisphere March mean SWE trend in the same period suggests an even larger decrease (~9 mm by unit conversion from their Fig. 1c).

The SWE trends in the various data sets have higher spatial correlations in Eurasia than in North America (last two columns of Table 5). However, there is a striking discrepancy in both the Eurasian and North American area mean 1981-to-2022 and 1981-to-2018 SWE trends between ERA5L and the other data sets (Table B1). ERA5L suggests an increase in average SWE in North America and a major decrease in Eurasia, while MERRA2, GlobSnow and CMIP6 MMM all indicate larger decreases in North America than Eurasia. In particular, GlobSnow shows a near-zero SWE trend in Eurasia but a 17 mm decrease from 1981 to 2018 in North America, in good agreement with Pulliainen et al. (2020). Mirroring these mean values, maps of the inter-data-set trend differences (Fig. B2) reveal a particularly pervasive difference between ERA5L and GlobSnow, with more negative trends in ERA5L in much of Eurasia but more positive trends in North America (Fig. B2c). Nonetheless, in the longer 1951-to-2022 period, the SWE decrease in ERA5L is also slightly larger in North America than in Eurasia (Table B1).

Regarding comparison to Kouki et al. (2022), the following text has been added to Section 4.

L263-272 / L285-294. The CMIP6 22-model mean SWE in the Total Snow Area is close to MERRA2 but 15 % below ERA5L; in the GlobSnow Area it is also below MERRA2. The average precipitation in the CMIP6 models exceeds both ERA5L and MERRA2, but this is compensated by lower mean values of F* and G (third row in Fig. 1). On the other hand, the average CMIP6 SWE exceeds the GlobSnow estimate (bottom-right corner in Fig. 1) by nearly 10 %. Kouki et al. (2022) also used GlobSnow as their main observational data set, finding an average ~15 % overestimate of March mean SWE for a larger set of 38 CMIP6 models (their Fig. 2b). Both the different sets of models and the inclusion of mountainous areas (where GlobSnow was replaced by other observational estimates) by Kouki et al. (2022) may contribute to this slight difference. Using linear regression, Kouki et al. (2022) attributed the overestimate of simulated SWE in February to too large November-to-January precipitation in the CMIP6 models. Although they made this regression analysis for February rather than March mean SWE, this result is in line with the CMIP6 MMM overestimate of area mean P* suggested by Fig. 1

Specific comments:

L11: 'This is repeated…' This sentence is a bit unclear, what is repeated?

Clarified as (L11 / L11): This trend attribution is repeated

L130: A sentence or two describing MERRA 2 might be good (as there is a short description of ERA5-Land).

This sentence has been addded (L135-136 / L142-143): MERRA2 is an atmosphere-land reanalysis produced by version 5.12.4 of the Goddard Earth Observing System atmospheric data assimilation system.

Table 1.: Possibly line numbers have jumped to the last column of the table? Or what are the numbers (145, 150) in the table?

Yes, they were line numbers, erroneously generated in the conversion from the Word document to a pdf file. Fortunately, all the tables in the revised manuscript are free of this problem.

Figure 3: It might be interesting to see the difference in SWE between different versions?

 Different spatial trends are visible in the figures but the same figures showing the differences might be beneficial.

These differences are shown in the new Figure B2.

 L520: Are there some areas where the spatial correlation is better? The average SWE values are larger in North America and some products have problems with this and thus have better performance in Eurasia.

I made the comparison separately for Eurasia and North America. In line with your speculation, the spatial correlations between the different data sets are better in Eurasia. However, there is a major discrepancy between the continental mean SWE trends between ERA5-Land and the other data sets. This is discussed in the last paragraph of Section 6 in the revised manuscript:

L400-408 / L450-458.The SWE trends in the various data sets have higher spatial correlations in Eurasia than in North America (last two columns of Table 5). However, there is a striking discrepancy in both the Eurasian and North American area mean 1981-to-2022 and 1981-to-2018 SWE trends between ERA5L and the other data sets (Table B1). ERA5L suggests an increase in average SWE in North America and a major decrease in Eurasia, while MERRA2, GlobSnow, and the CMIP6 MMM all indicate larger decreases in North America than Eurasia. In particular, GlobSnow shows a near-zero SWE trend in Eurasia, but a 17 mm decrease from 1981 to 2018 in North America, in good agreement with Pulliainen et al. (2020). Mirroring these mean values, maps of the inter-data-set trend differences (Fig. B2) reveal a particularly pervasive difference between ERA5L and GlobSnow, with more negative trends in ERA5L in much of Eurasia but more positive trends in North America (Fig. B2c). Yet, in the longer 1951-to-2022 period, the SWE decrease in ERA5L is also slightly larger in North America than in Eurasia (Table B1).

L521-523: "However, compared with both ERA5L and MERRA2..." I find this sentence bit unclear, what does overestimating the SWE decrease mean?

Reformulated as (L582-583 / L634-636): However, compared with both ERA5L and MERRA2, the CMIP6 models tend to simulate both larger SWE increases due to increasing precipitation and larger SWE decreases due to decreasing snowfall and snow-on-ground fractions.

**Response to Reviewer 2**

The author compares how three contributing factors that influence March historical snow water equivalent trends are represented in two reanalysis products and in 22 CMIP6 models. The factors contributing to total SWE change are changes related to the total precipitation, the fraction of precipitation occurring as snowfall, and the fraction of accumulated snowfall which remains on the ground. The author determines that there is broad agreement in the subcomponents over the long analysis period assessed (1951-2022) but that internal variability and model biases reduce the agreement between historical trend estimates and the CMIP6 multi-model mean over the shorter 1981-2022 period analyzed.

General Comments: Overall this paper is well-written and provides logically argued results. The framework used to attribute SWE changes to changes in precip, snowfall fraction, and snow-on-ground seems to works well and provide reasonable results. I only have a few issues I'd like to see addressed or commented on.

Specific comments:

140 The snow-on-ground term G absorbs a wide range of processes that will vary substantially from model to model. While inter-model difference in snow melt will be a key component of this term, differences in sublimation and any other structural differences in how a particular model treats snow will also affect it (e.g. vegetation-snow interactions, differences in how well snow water mass is locally conserved in a given model). Thus, it isn't surprising that the models disagree the most on the changes in this term. I think it's worth pointing this out in the text when you first discuss the equations and also at line 270 where you attribute the

145 differences in G primarily to snow melt.

The snow-on-ground fraction is indeed affected by a large number of factors. In response to this comment and related comments from Reviewer 3, the following paragraph was added to the revised manuscript:

L295-305 / L340-350. Table 4 also shows that the inter-model differences in G are in relative terms larger than those in P* and F*. This is perhaps unsurprising, since G may be affected by a multitude of factors. As

150 defined by Eq. (1), G reflects the balance between the source (accumulated snowfall) and sinks (snowmelt plus sublimation) of snow. The accumulated snowfall depends on both the amount and phase of precipitation, whereas snowmelt and sublimation are ultimately determined by the amount of energy that the land surface model allocates to them. The latter, in turn, is constrained by the downward solar and thermal radiation reaching the surface, the exchange of sensible and latent heat between the land surface

155 and the atmospheric models, the description of the surface albedo and emissivity, and the use or release of energy associated with temperature changes within the snow-ground-vegetation system. As many of these processes are described differently in different land surface models, it is perhaps unsurprising that the simulated SWE may vary substantially even between land surface models that share the same atmospheric forcing (Mudryk et al., 2015). A more detailed understanding of the causes of variation of G within the CMIP6

160 ensemble is an important target for future research.

Line 186: I believe you are defining X1=X_clim and X2 = X_clim + dx where dX is the monthly anomaly. Using these definitions x-bar is the monthly mean value plus half the monthly anomaly. If you actually used G_bar = G_clim + 0.5dG, F_bar = F_clim + 0.5dF, P_bar+0.5dP in equation (2) please state this more explicitly.

Yes, this is exactly what was done. I tried to clarify this by adding the following sentence to the description

165 of Equation (2) (L197 / L216-217): However, there is an implicit non-linearity in the coefficients $\bar{G}$, $\bar{F}$ and $\bar{P}$ in (2) since, for example, $\bar{G} = G_1 + \Delta G/2 \neq G_1$.

If you assumed the additional ½ dG/dF/dP components of G_bar/F_bar/P_bar would be absorbed into the non-linear term as second-order terms and approximated G_bar = G_clim, F_bar=F_clim, etc, please state this instead.

170 No, I did not do it this way.

I think the MMM trends you show are probably a good representation of the ensemble average given the range of models. But it would still be good to check that the figures look similar if you plot the median trend value at each location in case there is any undue influence in the mean from outlier trends.

I followed this suggestion. The trend maps are very similar; for example, the Total Snow Area spatial

175 correlation between the CMIP6 mean and median trends in SWE is 0.97 in both the 1951-to-2022 and 1981-to-2022 periods. I chose to not include this analysis in the manuscript but added the CMIP6 median values to Table 4 about the area mean statistics. These are indeed very similar with the multi-model mean values.

Minor comments:

L541: Please rephrase. I don't think snow melt occurs more efficiently as the climate warms. There's just
increased snowmelt in a warmer climate.

"Increased snowmelt" is also a slightly problematic formulation because, as integrated over the whole winter, the total amount of snowmelt is the same as the accumulated snowfall (which may either increase or decrease). Therefore, this was reformulated as (L601 / L654-655) "enhanced snowmelt in a warmer climate".

**Response to Reviewer 3**

The manuscript "Changes in March mean snow water equivalent since the mid-twentieth century and the contributing factors in reanalyses and CMIP6 climate models" uses a decomposition-based approach to evaluate and diagnose changes in historic snow water equivalent (SWE) trends in the northern hemisphere for two historic periods in two reanalysis products, the GlobSnow product, and the CMIP6 ensemble. SWE is decomposed into contributions from total precipitation, snow fraction, and snow-on-ground fractions. The primary findings include high degrees of spatial correlations but individual variation due to model-to-model differences in snow-on-ground fraction. Total SWE has gone up in many far northern locations due to increases in precipitation, though CMIP6 appears to overestimate SWE increases. Biases in all data sources are discussed, and the paper highlights how difficult SWE is to correctly simulate in climate models. The topic is within the scope of The Cryosphere.

Overall, I like the methodology as an interesting approach to breaking down drivers of SWE. However, my first major concern centers around choices in the spatial aggregation step and the monthly time step that may lead to very different results. Huge efforts and computational resources have been expended on improving spatial resolution of snow (and many key climate system variables) in global models, and it is unfortunate to not see how finer resolution may improve the insight provided by the work (it also may not, but it needs to be shown). My second major concern is the lack of energy balance components in SWE trends is perhaps implicit in the G term, but this is not apparent from the methodology description.

My response to these comments is divided to three parts, labeled as (1)-(3) below.

(1) I start from the G term because this explanation may also shed light on why the time resolution is not important. In this study, G is a purely diagnostic quantity. From Eq. (1) one gets

$$G = \frac{SWE}{\int_{t_0}^{t} FP dt'} = \frac{SWE}{\int_{t_0}^{t} Snowfall\ dt'} = \frac{SWE}{accumulated\ snowfall}$$

For calculating $G$, one only needs the SWE and the accumulated (i.e., time-integrated) snowfall from the beginning of the winter season. Both the SWE and the snowfall are directly available from reanalysis and climate model output. To clarify, the paragraph preceding Eq. (1) was rewritten as:

L177-180 / L196-199. Our diagnostic framework follows Räisänen (2008, 2021a). Only three variables are needed from a reanalysis or a model simulation: monthly means of SWE, snowfall, and total precipitation (P). The monthly snowfall is first rewritten as $FP$, where $F$ is the fraction of precipitation that falls as snow. SWE in month $t$ then becomes

$$SWE = G \int_{t_0}^{t} FP dt' \tag{1}$$

Clearly, there are many different processes and meteorological factors that affect G. A detailed analysis of these factors in the individual reanalyses and climate models is well beyond of what can be reasonably included in the current paper. However, I have pointed out the importance of this issue by adding the following paragraph to the end of Section 4 in the revised manuscript:

L295-305 / L340-350. Table 4 also shows that the inter-model differences in G are in relative terms larger than those in P* and F*. This is perhaps unsurprising, since G may be affected by a multitude of factors. As defined by Eq. (1), G reflects the balance between the source (accumulated snowfall) and sinks (snowmelt plus sublimation) of snow. The accumulated snowfall depends on both the amount and phase of precipitation, whereas snowmelt and sublimation are ultimately determined by the amount of energy that the land surface model allocates to them. The latter, in turn, is constrained by the downward solar and thermal radiation reaching the surface, the exchange of sensible and latent heat between the land surface and the atmospheric models, the description of the surface albedo and emissivity, and the use or release of energy associated with temperature changes within the snow-ground-vegetation system. As many of these processes are described differently in different land surface models, it is perhaps unsurprising that the simulated SWE may vary substantially even between land surface models that share the same atmospheric forcing (Mudryk et al., 2015). A more detailed understanding of the causes of variation of G within the CMIP6 ensemble is an important target for future research.

(2) The low spatial resolution of the analysis leads to a loss of potentially valuable local information from the high-resolution analysis products. However, the 2.5° × 2.5° grid is sufficient when the focus is on large-scale trends, as is the case in this paper. Importantly, the trend results in this coarser grid are insensitive to the order between grid remapping and the decomposition (Eqs. (1)-(2) in the manuscript). In other words, if one starts with remapping all the required input data to the 2.5° grid before applying Eqs. (1)-(2), the resulting trends are very similar to those obtained by first doing the calculations in a finer grid and then remapping the trends to the 2.5° grid. The only (expected) exception are 2.5° grid boxes that are partly covered by sea in the original grid, because the required input data are only available over land. This similarity, as well as the loss of small-scale information, is illustrated in the new Figure B1.

In any case, redoing the analysis on a finer resolution would have a low benefit-to-effort ratio, because I have only archived a large fraction of my CMIP6 data in the 2.5° grid. Furthermore, as most of the CMIP6 models only have a resolution of 1-2°, using a grid finer than 2.5° would not help to draw much new information from the models.

(3) The choice to use monthly rather than daily data in the analysis is also a pragmatic one. Using daily data would increase both the data volume and the computing time by a factor of 30. In principle, it would be better to do the calculation with daily than monthly mean data, because this would allow a more precise evaluation of the time-integrated snowfall in Equation (1). However, in practice, this difference is very unlikely to be important.

To get an idea of the potential impact of the time resolution of the input data, I recalculated the 1951-to-2022 trends for ERA5-Land using an even coarser time resolution: 2 months instead of 1 month. Constrained by the 2-month resolution, this analysis was done for the trends in the February-March mean SWE rather than March mean SWE. In Figure R1 below, the results are compared to the corresponding February-March mean trends calculated with 1-month time resolution.

[Figure]

**Figure R1.** Decomposition of 1951-to-2022 trends in February-March mean SWE, as calculated by (top) monthly and (middle) 2-monthly ERA5-Land data. The difference is shown in the bottom row, using factor of 5 smaller intervals in the color scale.

The results with 1- and 2-month time resolution are very similar. The spatial correlations for the trends in $\Delta SWE(\Delta P)$, $\Delta SWE(\Delta F)$, $\Delta SWE(\Delta G)$ and SWE are 0.99, 0.96, 0.95 and 0.99, respectively. The reason why the SWE trends are not identical is the subtraction of the SWE in the first time unit (August for 1-month but August-September for 2-month time resolution) when calculating the "seasonal component" of the February-March mean SWE. Generally, the time resolution mainly affects the trend decomposition results in those areas where non-negligible snowfall already occurs in September. Elsewhere, the differences are negligible.

Because climatic conditions such as the amount of snowfall change more between two months than within a single month, this comparison likely gives an upper estimate of the errors that monthly time resolution may lead to.

The paper is written well. For the audience I expect to find this paper of the most benefit (e.g., modeling and observational groups focused on snow processes), the figures are satisfactory. There are numerous occasions where qualitative information is given (e.g., L232 "slightly larger") or implied (L235 "exceeds") when numerical values would be welcomed as they are more informative.

There is a delicate balance between the exactness and the readability of the text. I have added several more numerical values in the revised manuscript text, where I found this appropriate. In other cases, I have added more explicit references to the figures or tables from which the values can be seen. Still in other cases it would be potentially misleading to try to give a precise number or even a range of values, particularly when referring to features seen in maps (of this article or earlier ones). Since verbal definitions of areas (e.g., "easternmost Siberia") will be interpreted differently by different persons, the meaning of seemingly well-defined numerical values would also be interpreted differently.

The paper would also benefit from additional citations to shore up arguments and to increase its comprehensive coverage of the topic. With revisions to address the major, minor, and specific comments, I believe this manuscript will make a solid contribution to the literature.

Many additional citations have been added. I apologize if (as is likely) I have missed something important, but sometimes it is surprisingly difficult to find good references to results that appear to be established "silent knowledge" in the community. Also, as the paper is quite long already, making it even longer with a comprehensive discussion of earlier literature would make it less attractive to many readers.

Major Comments:

310     1.  The aggregation via interpolation of observational data and CMIP6 simulations to 2.5° horizontal resolution likely will cause a lot of loss of potentially valuable information, especially in mountainous areas where SWE can vary over much smaller spatial scales. I am concerned that the results and interpretations may be very different if different resolutions are used. There may be good reason to do this aggregation as performed, but this needs to be shown. In other words, how much do the
315     results vary if the ERA5, MERRA2, CMIP6, and GlobSnow are all re-gridded to 50 km instead of such a coarse 2.5° resolution? This exercise may also highlight where models are doing better/worse across locations in the Northern Hemisphere and be valuable to modeling groups who want to improve their results.

320     See above. I agree that a trend analysis on a finer grid would give valuable insights on regional SWE change dynamics in (e.g.) many mountain areas, but this is a topic for another study. The focus in this study is on large-scale features, the 2.5° resolution is sufficient for this, and the trend decomposition results on this scale would not be materially altered even if higher-resolution data were used before re-gridding (rows 2-3 in the new Figure B1). Also note that the resolution for some of the CMIP6 models is close to 2.5°, so no higher-
325     resolution information is available for them.

        1.  What physical components control the SWE on the ground (the G term in eq 1)? Does this term accumulate the effects of the surface energy budget? More explanation is needed as this is a critical component of SWE trends (the "demand side" whereas F and P are the "supply side"). While the
330         focus on the supply side of SWE trends is valuable, adding an analysis of G, especially if broken down into its physical components, would greatly elevate the papers contribution to understanding what is driving SWE trends as well as be helpful for the modeling community to improve the representation of these key processes (L270 and L349 acknowledges the challenges of snowmelt modeling and role of how this step leads to differences in results). This would allow the paper to nicely 'bookend' the
335         supply and demand sides in the CMIP6/historic reanalysis framework.

        See above. This study only attempts to answer how trends in G have affected SWE. Diagnosing why G has changed in the way that it has would require very extensive additional analysis. This is far beyond the scope of the present manuscript. Nevertheless, a brief discussion of the factors that may affect G has been added
340     to the end of Section 4 (L295-305 / L340-350, see the red text copied to P6 of this response letter).

        Minor Comments:
        1.  It was unclear what timestep of observational and model output was used, was it monthly March means or the date of peak snow water equivalent in March (daily). Similar to Major Comment #1,
345         how sensitive are the results to the use of monthly versus daily values? After reading further, it appears March monthly mean was used, but was this necessary for some reason? Why not use the peak March value (which likely would vary in time across the month by latitude, elevation, a given year, etc)?

350     This was already discussed above. The choice of monthly time resolution was largely pragmatic. To be more explicit about the time resolution of the input data, the following sentence has been added (L177-178 / L196-197): Only three variables are needed from a reanalysis or a model simulation: monthly means of SWE, snowfall, and total precipitation (P).

355     1.  L229: This difference in mean SWE could be readily examined by using finer (native) resolutions, making a stronger point than the speculation provided at present.

Yes, it might be possible to analyze the values of SWE together with model orography in major mountain regions to get more insight on this. However, I feel that this is out of the main focus of the study, and have therefore not pursued this further.

360

    1. It would help to have a bit more comprehensive literature review on SWE comparisons between reanalysis products such as MERRA2 and ERA5L across locations and scales; this may also be helpful when discussing the results found.

365

Although there are several studies in which SWE in different data sets is compared, the most salient information condenses on two points: (1) how large is the overall variability between data sets and (2) what can be said about the likely bias in ERA5-Land, MERRA2 and the CMIP6 models relative to the "real" real-world SWE. Thus, the revised manuscript will include two additions in this purpose. First, the following

370 sentences have been added to the Introduction:

(L79-82 / L79-82). On the other hand, model biases and limitations in the observational input result in a large spread between the SWE estimates from different analysis products (Mudryk et al., 2015; Mortimer et al., 2020). Mudryk et al. (2015) found more than a factor of 1.5 range even in the Northern Hemisphere total

375 winter peak snow mass among the five datasets that they evaluated (their Fig. 3a).

Second, the following paragraph has been added to the middle of Section 4, where the discussion is about whether the CMIP6 models over- or underestimate the average SWE:

380 L274-281 / L296-306. Assuming that GlobSnow and the other observational estimates used by Kouki et al. (2022) are correct, the average SWE is too large in both the CMIP6 MMM, MERRA2, and (especially) ERA5L. A comparison of GlobSnow with three other bias-corrected estimates of the total snow mass in Northern Hemisphere non-alpine areas (Table 1 of Pulliainen et al., 2020) supports this assessment: all the four estimates are within 7.4 %, GlobSnow being the highest. However, SWE in mountainous areas is known less

385 well and may be severely underestimated in many gridded analyses (Snauffer et al., 2016; Wrzesien et al., 2018). Despite the higher mean SWE in ERA5L than in MERRA2 and GlobSnow, Muñoz-Sabater et al. (2021) found ERA5L to underestimate SWE by ~50 % at the five Earth System Model – Snow Model Intercomparison Project (Krinner et al., 2018) alpine reference sites.

390     1. I liked the two directions posed in the conclusion, however the way the conclusion was organized, the paper felt like it ended abruptly. I suggest adding a short closing section to finish the article or reorganizing this section.

Again, this is a good comment. Re-reading the original manuscript, I realized that it missed a discussion point

395 on what stakeholders should do with the SWE projections, and the research needs related to this. This text was therefore added to the end of the Conclusions section:

L619-632 / L673-684. Another important question is, how water resource managers and other stakeholders needing SWE projections should use the information available from climate model ensembles. On the face

400 of it, the answer seems disappointing. As real-world SWE trends appear to have hitherto been further away from the model-simulated trends than the latter are from each other, this might also well apply to the trends in the future. It would therefore seem prudent to allow for all the uncertainty suggested by the variation between climate models, possibly adding a safety margin for systematic model errors. However, this conclusion may need re-evaluation if future research is able to reduce the uncertainty in the observed trends.

405 Moreover, the practice of giving all models the same weight in projections of future SWE change is probably sub-optimal, particularly for longer-term projections in which model differences grow increasingly dominant over internal variability. For example, Räisänen (2008) found a dependence between model-simulated

baseline winter temperatures and projected future SWE changes, which makes increases of SWE more likely in models with a cold than a warm temperature bias. He also used inter-model cross validation to show that this physically expected dependence could be potentially used for improving probabilistic projections of SWE change. Furthermore, inter-model variations in the Northern Hemisphere snow albedo feedback are strongly correlated between seasonal and climate change time scales (Hall and Qu, 2006; Qu and Hall, 2014). This does not guarantee that a similar cross-time-scale constraint would be available for SWE as well, but it suggests that the possibility is worth exploring.

Specific Comments:
L23-32: Suggest to add citations to the first paragraph; all good info but offers opportunity to point readers to recent work on various sub-topics.

Several new citations have been added to the first paragraph in the Introduction (L23-33 / L23-33). I would have been pleased to find even more, but it turned out to be surprisingly difficult to find good references even to some seemingly well-known results. Part of the difficulty is that the literature on model-simulated changes in snow conditions (which is the theme of this paragraph) is more limited than that on observed changes.

L41: Add "beneficial" before "ecological impacts"?

Suggestion accepted (L42 / L42).

L45-54: This paragraph would be stronger with additional numerical quantification of historic change and more local specifics, e.g., what are the decreasing trends?

I found two numerical values that are sufficiently easy to understand and well-defined enough for addition to this paragraph:

L49-52 / L49-52. Pulliainen et al. (2020) evaluated trends in snow mass in 1980-2018 using the Global Snow Monitoring for Climate Research (GlobSnow) v3.0 analysis. Focusing on non-mountainous areas north of 40ºN, they found a statistically significant decreasing trend in March mean snow mass in North America (best estimate: -4 % decade$^{-1}$) but a near-zero trend in Eurasia.

… L55-58 / L55-58. Using observations for the years 1981-2010, Mudryk et al. (2017) found the monthly mean snow cover extent in the extratropical Northern Hemisphere land areas to decrease throughout the snow season by $(1.9 \pm 0.9) \times 10^6$ km$^2$ for each 1°C increase in the mean temperature in the same month and area.

L69: Suggest removing "three-dimensional" as weather prediction and climate models are four-dimension (3-d in space plus a time dimension). Would also benefit to add some examples of these to give readers examples of their spatial and temporal resolution of output.

The word "three-dimensional" has been removed (L73 / L73), and the following example of spatial and temporal resolution has been added (L77-79 / L77-79): Reanalyses and land surface models produce spatially complete SWE simulations, in some cases with high spatial and temporal resolution (e.g., hourly data at 9 km resolution for ERA5-Land (Muñoz Sabater et al., 2021)).

L81: add the scale this paper addresses (Northern Hemisphere) to "recent trends in SWE" just to set the stage for going into the questions.

Suggestion accepted (L89 / L89).

L107-112: Instead of this table-of-contents explanation of upcoming sections, I think this text could be markedly condensed and some initial insights into what the work provides the community could be given, thus helping motivate the work better.

Table of contents was replaced with this sentence (L114-117 / L115-118): A key finding of this research is a reasonable agreement between the ERA5-Land reanalysis and the CMIP6 (6th phase of the Coupled Model Intercomparison Project; Eyring et al., 2016) models on the March mean SWE trends and their contributing

factors in the period 1951-2022 (Section 5) but a worse agreement between various observational data sets with both each other and the CMIP6 models on the trends from 1981 to 2022 (Section 6).

L138: Please quantify "thick"
L138: How much is this underestimation of SWE?

The essence of the problem is that the microwave signal saturates when SWE reaches ca. 150 mm, and the uncorrected GlobSnow SWE ceases to increase when this limit is exceeded. The corresponding sentence was reformulated as (L142-144 / L149-151): These corrections are based on comparison with snow course measurements, and they improve the SWE estimates especially in areas with thick snow, where the noncorrected data systematically underestimate SWE due to the saturation of the microwave signal when SWE exceeds ca. 150 mm (Pulliainen et al., 2020).

L141, 167: Tables 1 and 2: Please add a column of the native spatial horizontal resolution of these products

A column of the native horizontal resolution has been added to Tables 1 and 2.

L154: Please define the SSP scenarios briefly and add a reference.

SSP = Shared Socioeconomical Pathways, and the chosen SSP2-4.5 scenario is often characterized as a

"middle-of-the-road" future, although the difference from other scenarios only becomes important in the longer-term future. The sentence has been rewritten as (L153-155 / L168-171): Climate model simulations from CMIP6 were also used, concatenating historical simulations for the years 1950-2014 with simulations for the Shared Socioeconomical Pathways "middle of the road" 2-4.5 scenario (Fricko et al., 2017) for years 2015-2022.

L165: Please add a reference for conservative remapping (even if just the function used in the programming language) to ensure repeatability.

A reference has been added on L171 / L190 to:

Jones, P. W.: First- and second-order conservative remapping schemes for grids in spherical coordinates, Mon. Weather Rev., 127, 2204-2210, https://doi.org/10.1175/1520-0493(1999)127<2204:FASOCR>2.0.CO;2, 1999.

L172: Table 2: suggest making the delta-T subscripts hyphenated (51-22) to be more intuitive.

Suggestion accepted (Table 2).

L180: What fraction of areas in the Northern Hemisphere have permanent snow cover? Has this number changed substantially? It would be OK to give the mean or median value for a given date range to give readers an idea even if this is calculated for each specific year.

I chose to answer a slightly modified question that is more important for the current work: in how large a fraction of the analysis area does the choice to subtract the August mean SWE affect the results?

L184-188 / L203-207. To make Eq. (1) also applicable in areas where snow cover regularly or sporadically survives to the late summer, we subtract the August mean SWE from the left-hand-side, thus focusing on the seasonal component of SWE. For reference, in ERA5L about 7 % of the Total Snow Area (red and yellow shading in the bottom-left panel of Fig. 1) has non-negligible (> 5 mm) time mean August SWE in the 2.5° × 2.5° grid, largely in mountainous and Artic areas. For MERRA2, this fraction is only 1 %.

Typically: two orders of magnitude smaller than the other terms (L195-196 / L214-216): Thus, the anomaly in SWE is decomposed to contributions from the total precipitation ($\Delta P$), snowfall fraction ($\Delta F$) and snow-on-ground fraction anomalies ($\Delta G$), plus a non-linear term that is typically two orders of magnitude smaller than the others.

L239: Alternatively, could the ERA5L SWE being "too large" be a product of it doing a better representation of SWE at high elevations as well?

Yes, it could. The following paragraph was added to the middle of Section 4, to acknowledge among other things that the SWE in mountainous areas is not well known:

L274-281 / L296-306. Assuming that GlobSnow and the other observational estimates used by Kouki et al. (2022) are correct, the average SWE is too large in both the CMIP6 MMM, MERRA2, and (especially) ERA5L. A comparison of GlobSnow with three other bias-corrected estimates of the total snow mass in Northern Hemisphere non-alpine areas (Table 1 of Pulliainen et al., 2020) supports this assessment: all the four estimates are within 7.4 %, GlobSnow being the highest. However, SWE in mountainous areas is known less well and may be severely underestimated in many gridded analyses (Snauffer et al., 2016; Wrzesien et al., 2018). Despite the higher mean SWE in ERA5L than in MERRA2 and GlobSnow, Muñoz-Sabater et al. (2021) found ERA5L to underestimate SWE by ~50 % at the five Earth System Model – Snow Model Intercomparison Project (Krinner et al., 2018) alpine reference sites.

Corrected (L247 / L268).

L281: Suggest to be more specific in terms of temperature (and radiation), also add citation(s) here unless this finding is directly shown, in which case please reference the figure or table here.

The sentence that this comment refers to ("This term is the most negative in mid-latitude North America and in a zone extending from eastern Europe to southern Scandinavia, where the main snowmelt season is ongoing in March and has been advanced by rising spring temperatures.") is based partly on analysis of the ERA5-Land data, partly on my knowledge of the climate in the mentioned areas. I could not find a good earlier reference to this result, but I find it physically intuitive. Also note that, although seasonal increases in both temperature and solar radiation contribute to the spring snowmelt, the greenhouse gas induced climate change is expected to affect temperature much more than the insolation. Thus, no changes in the text.

L284-285: Add references; also would help to provide some additional physical reasoning about the logic involved in this sentence.

The reasoning is that, as far as the mean temperature is well below zero, the frequency of above-zero temperatures that lead to significant snowmelt remains small (L317-319 / L364-366): Although warming is generally expected to enhance snowmelt, this effect is modest where the mean temperature in March and in the preceding winter months is well below zero, so that above-zero temperatures remain uncommon despite the warming (Räisänen, 2008).

L317-320: Would help to give trend values in the text for easy comparison.

The trend values depend on the precise delineation of the areas, which is necessarily diffuse. Therefore, I just added a reference to the appropriate figure panels in this sentence (L353-355 / L402-404): For example, increases in SWE in northern Eurasia in ERA5L are much more widespread in 1951-2022 than in 1981-2022, except for easternmost Siberia, where SWE decreases in the former period but increases in the latter (top-right panels in Figs. 2 and 3).

L331-334: Does this suggest the models are getting temperature signals correct? I see later this is noted in a few paragraphs (L343); perhaps consider reorganizing the text in this section and the other to allow it to flow more naturally?

I considered the reorganization, but it would create another problem because the finding also applies to the CMIP6 simulations that are only discussed after this sentence. Therefore, I chose to add a brief explanation in this sentence ("as argued below …") and keep the full discussion where it was.

L367-369/ L416-418.  Thus, ERA5L and MERRA2 agree reasonably well on the SWE trend caused by changing snowfall fraction (which, as argued below, is a relatively straightforward response to warming), but much less well on the trends associated with changes in total precipitation and the snow-on-ground fraction.

L370: change "Both" to lowercase

Corrected.

L371: all or nearly all, can you rephrase to give a specific number?

Rewritten as (L424-426 / L474-476): In MERRA2, changes in total precipitation make a smaller positive contribution to the SWE trend since winter 1981 than any of the models simulate, and the trend in ERA5L is also exceeded in 20 of the 22 models (Table 4).

L396: Change to: "With regards to the components…" or "Regarding the components"

Changed to "Regarding" (L451 / L501).

L428: To what extent might relative humidity (or direct moisture variables) be at play in the snowfall fraction? There has been a lot of recent work highlighting the importance of atmospheric moisture in determining precipitation phase (see e.g., Jennings et al. 2018 Nature Communications). Could be worth mentioning here.

It is true that both the relative humidity and the lower tropospheric lapse rate also affect the phase of precipitation. However, in the context of climate change, changes in these two are very likely to be less

important than temperature change. I therefore decided to include these additional factors only in an implicit way, by adding two references in the following sentence that appears slightly earlier in the text (L380-383 / L429-433): This is most likely (i) because the phase of precipitation is primarily (Auer, 1974) although not completely (Sims and Liu, 2015; Jennings et al., 2018) determined by temperature and (ii) because the observational uncertainty is smaller (Gulev et al., 2021) and the signal-to-noise ratio is higher (Räisänen, 2001; Hawkins and Sutton 2009, 2011; Lehner et al., 2020) for temperature than precipitation changes.

L465: Is this comparison of spatial correlation performed at 2.5° resolution or at finer scales?

The revised text explicitly mentions that the correlations were calculated at the 2.5° resolution (L521-523 / L573-575): For example, the spatial correlation between the CRU and ERA5L trends in the 2.5° grid is 0.69 for temperature but only 0.39 for precipitation (Table B2).

L474: This is interesting discussion; have other authors suggested, found similar results, or further investigating these aspects in CMIPx? Worth adding if so.

Yes. Lehner et al. (2020) confirm these results for the more recent CMIP5 and CMIP6 simulations, and I therefore added this reference to L485 / L536:

Lehner, F., Deser, C., Maher, N., Marotzke, J., Fischer, E. M., Brunner, L., Knutti, R., and Hawkins, E.: Partitioning climate projection uncertainty with multiple large ensembles and CMIP5/6, Earth Syst. Dynam., 11, 491–508, https://doi.org/10.5194/esd-11-491-2020, 2020.

L501: Instead of "Height" would "maximum extent" or "maximum depth" be better terms?

Rewritten as (L558-559 / L610-611): The focus was on SWE in March, when the Northern Hemisphere snow mass is the largest (Pulliainen et al., 2020).

L513: Here's a great example of where numerical values would help readers and would be useful for citing this work! How much compensation of increased SWE from precip is there in the other direction (loss) from FP and G declines? OK to give an average and the range (e.g., 1 standard deviation).

Average values for ERA5-Land were added. Further details would make the text too complicated.

L569-572 / L621-624. **Trends from winter 1951 to 2022.** ERA5L and the CMIP6 models agree qualitatively well on the dynamics of SWE change. Although increasing total precipitation has acted to increase SWE in most of the extratropical Northern Hemisphere (average contribution in ERA5L in the Total Snow Area: 7.1 mm), this has been more than compensated by reduced snowfall fraction (-8.0 mm) and, in most areas, reduced snow-on-ground fraction (-7.4 mm).

L540: Per Major Comment 2, did this approach address how snowmelt occurs and what drives it becoming more rapid or frequent as the climate warms (I'm not sure "efficient" is the right word here, either, though I suppose if SWE is broken down into melt and sublimation terms, water input to the surface from snowmelt would be less efficient per unit SWE if more water was lost to sublimation).

More efficient has been replaced by enhanced snowmelt in a warmer climate (L601 / L654). However, as discussed above, a more detailed discussion of the drivers of the decreasing snow-on-ground fraction is beyond the scope of the current analysis, which simply diagnoses this fraction from the ratio between SWE and accumulated snowfall.